# RETHINKING DRIVING WORLD MODEL AS SYNTHETIC DATA GENERATOR FOR PERCEPTION TASKS

**Kai Zeng**[1,2*]**, Zhanqian Wu**[2*]**, Kaixin Xiong**[2] **, Xiaobao Wei**[1,2†]**, Xiangyu Guo**[2,3] **,
Zhenxin Zhu**[2] **, Kalok Ho**[2] **, Lijun Zhou**[2] **, Bohan Zeng**[1] **, Ming Lu**[1] **,
Haiyang Sun**[2†]**, Bing Wang**[2] **, Guang Chen**[2] **, Hangjun Ye**[2‡]**, Wentao Zhang**[1,4,5‡]
[1] Peking University    [2] Xiaomi EV    [3] Huazhong University of Science and Technology
[4] Beijing Key Laboratory of Data Intelligence and Security (Peking University)
[5] Zhongguancun Academy

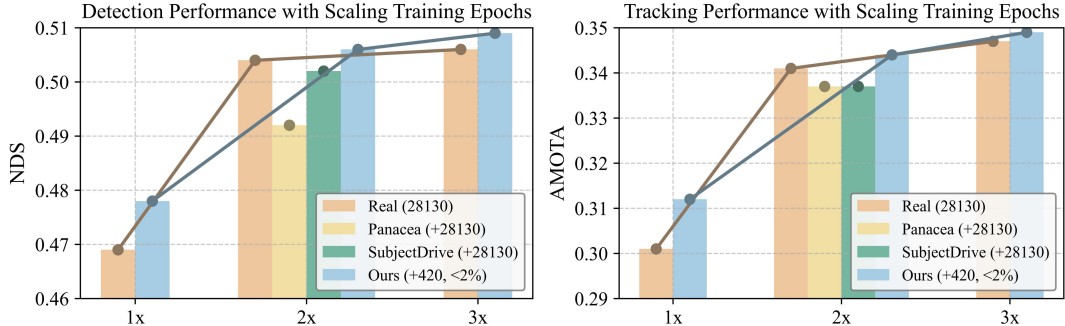

Figure 1: Dream4Drive demonstrates the effectiveness of synthetic data: with fewer than 2% synthetic samples, it consistently improves detection and tracking across epochs, outperforming previous data augmentation baselines under **fair evaluation**. 1× denotes the baseline training epochs; 2× and 3× represent twofold and threefold increases, respectively.

## ABSTRACT

Recent advancements in driving world models enable controllable generation of high-quality RGB videos or multimodal videos. Existing methods primarily focus on metrics related to generation quality and controllability. However, they often overlook the evaluation of downstream perception tasks, which are **really crucial** for the performance of autonomous driving. Existing methods usually leverage a training strategy that first pretrains on synthetic data and finetunes on real data, resulting in twice the epochs compared to the baseline (real data only). When we double the epochs in the baseline, the benefit of synthetic data becomes negligible. To thoroughly demonstrate the benefit of synthetic data, we introduce Dream4Drive, a novel synthetic data generation framework designed for enhancing the downstream perception tasks. Dream4Drive first decomposes the input video into several 3D-aware guidance maps and subsequently renders the 3D assets onto these guidance maps. Finally, the driving world model is fine-tuned to produce the edited, multi-view photorealistic videos, which can be used to train the downstream perception models. Dream4Drive enables unprecedented flexibility in generating multi-view corner cases at scale, significantly boosting corner case perception in autonomous driving. To facilitate future research, we also contribute a large-scale 3D asset dataset named DriveObj3D, covering the typical categories in driving scenarios and enabling diverse 3D-aware video editing. We conduct comprehensive experiments to show that Dream4Drive can effectively boost the performance of downstream perception models under various training epochs. Project website: https://wm-research.github.io/Dream4Drive/.

*Equal Contribution. Email: `asbeforekz@gmail.com`
†Project Leader.
‡Equal Corresponding Author. Email: `wentao.zhang@pku.edu.cn`

# 1 INTRODUCTION

Perception tasks such as 3D object detection (Li et al., 2022; 2024b; Wang et al., 2023a; 2025) and 3D tracking (Wang et al., 2023a; Zhang et al., 2023c; Han et al., 2025; Li et al., 2025b), which support planning and decision-making (Jiang et al., 2023; Hu et al., 2023; Shan et al., 2025; Hao et al., 2024), are extremely important in autonomous driving. The performance of perception models, however, is highly dependent on large-scale annotated training datasets (Caesar et al., 2020; Wang et al., 2023b). To ensure reliability in rare but critical safety scenarios, it is essential to gather adequate long-tail data. Although the autonomous driving community has developed a thorough 3D annotation pipeline to facilitate data acquisition (Zhao et al., 2025b), collecting long-tail data remains highly time-consuming and labor-intensive.

Driving world models based on diffusion and ControlNet (Rombach et al., 2022; Zhang et al., 2023b) generate synthetic data from scene layouts and text (Gao et al., 2023; Wen et al., 2024; Guo et al., 2025), but offer limited control over object pose and appearance, reducing data diversity (Li et al., 2025a). Editing-based methods (Singh et al., 2024; Liang et al., 2025b; Yu et al., 2025) improve this by inserting objects using reference images and 3D boxes, yet their single-view nature restricts use in multi-view BEV perception. Reconstruction-based approaches (NeRF, 3DGS) (Chen et al., 2021; Zanjani et al., 2025) provide geometric control but suffer from artifacts due to sparse views and lack illumination modeling, causing inconsistencies between inserted objects and the background.

More importantly, we argue that the data augmentation experiments of previous methods (Wen et al., 2024; Li et al., 2024a) are unfair, as they usually leverage a training strategy that first pretrains on synthetic data and finetunes on real data, resulting in twice the epochs compared to the baseline (real data only). We find that, under the same number of training epochs, large amounts of synthetic datasets offer little to no advantage and can even perform worse than using real data alone. As shown in Fig. 1, under the 2× epoch setting, models trained exclusively on real data achieve higher mAP and NDS compared to those trained on real and synthetic data. Given the importance of downstream perception tasks for autonomous driving, we believe it is essential to rethink the effectiveness of the driving world model as a synthetic data generator for these tasks.

To reevaluate the value of synthetic data, we introduce Dream4Drive, a novel 3D-aware synthetic data generation framework designed for downstream perception tasks. The core idea of Dream4Drive is to first decompose the input video into several 3D-aware guidance maps and subsequently render the 3D assets onto these guidance maps. Finally, the driving world model is fine-tuned to produce the edited, multi-view photorealistic videos, which can be used to train the downstream perception models. Consequently, we can incorporate various assets with different trajectories(e.g., views, poses, and distance) into the same scene, significantly improving the diversity of the synthetic data. As shown in Fig. 1, under identical training epochs (1×, 2×, or 3×), our method requires only 420 synthetic samples—less than 2% of real samples—to surpass prior augmentation methods. To be best of our knowledge, we are the first to demonstrate under fair comparisons that synthetic data can provide real benefits beyond training solely on real data.

Specifically, Dream4Drive leverages a multi-view video inpainting model finetuned from the Diffusion Transformer (Peebles & Xie, 2023). Unlike prior methods that rely on sparse spatial controls (e.g., BEV maps and 3D bounding boxes), Dream4Drive uses dense 3D-aware guidance maps (Yu & Smith, 2019; Liang et al., 2025a) (e.g., depth, normal, edge, cutout, and mask) to preserve the geometry and appearance of the original video, while editing them by rendering 3D assets into these maps. This design enables instance-level, cross-view consistent video editing, ensuring both visual realism and geometric fidelity. The generated videos not only achieve superior quality but can also be directly used to train state-of-the-art perception models (Wang et al., 2023a).

To facilitate diverse 3D-aware video editing, we design a pipeline that automatically acquires high-quality 3D assets based on a target scene image or video of a desired asset category. We first apply the image segmentation model (Ren et al., 2024; Lin et al., 2024) to localize and crop objects of the specified category, then employ the image generation model (Wu et al., 2025) to generate multi-view consistent images of the target object. These images are fed into a mesh generation model (Hunyuan3D et al., 2025) to generate high-quality 3D assets. We present DriveObj3D, a large-scale 3D asset dataset that encompasses typical categories found in driving scenarios, to support future research. Our main contributions are:

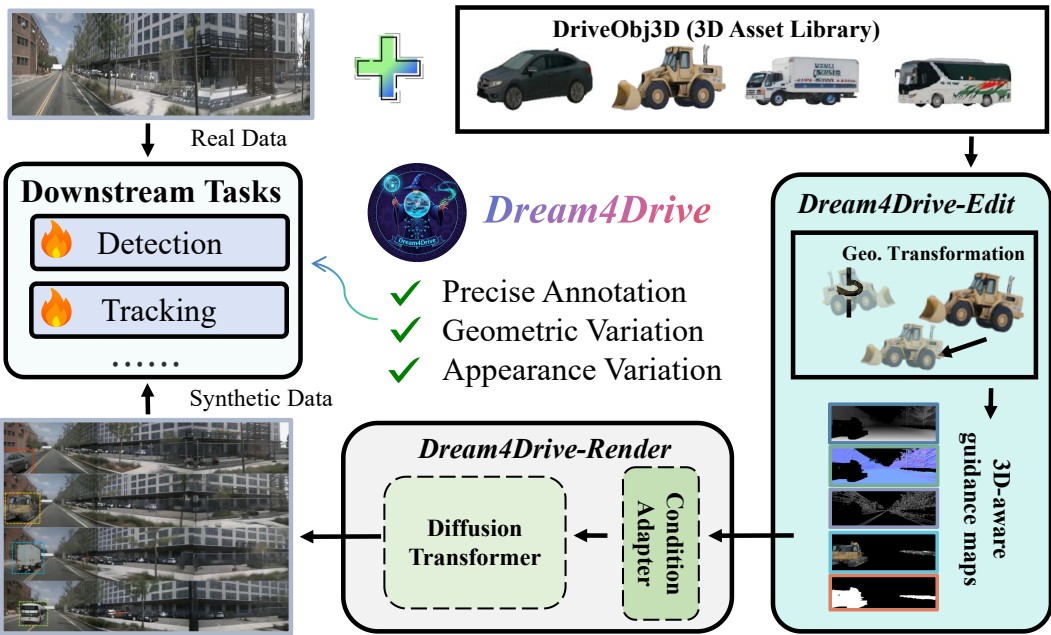

Figure 2: The illustration of **Dream4Drive**, which provides precise annotations, geometric variety, and appearance diversity to improve downstream perception tasks.

- We find that previous data augmentation methods are evaluated unfairly: under the same training epochs, hybrid datasets do not show any advantage over real data alone.
- We propose Dream4Drive, a 3D-aware synthetic data generation framework that edits the video with dense guidance maps, producing synthetic data with diverse appearances and geometric consistency.
- We contribute a large-scale dataset named DriveObj3D, covering the typical categories in driving scenarios for 3D-aware video editing.
- Extensive experiments across different training epochs show that adding less than 2% synthetic data can significantly improve perception performance, highlighting the effectiveness of Dream4Drive.

## 2 RELATED WORK

**Video Generation in Autonomous Driving.** High-quality data is crucial for training perception models in autonomous driving, motivating growing interest in driving video generation. Early approaches (Yang et al., 2023; Gao et al., 2023; Wen et al., 2024) employ diffusion models with ControlNet, conditioned on BEV maps (Ma et al., 2024b) and 3D bounding boxes, to generate paired image data (Zhou et al., 2024; An et al., 2024; 2025b). Recent works (Gao et al., 2024a; Jiang et al., 2024; Li et al., 2024a; Ji et al., 2025; An et al., 2026; Li et al., 2025c) adopt powerful Diffusion Transformers (DiTs) to further enhance generation quality. SubjectDrive (Huang et al., 2024a) uses an external subject bank to enhance vehicle appearance diversity. However, these methods depend on original scene layouts, which limit geometric diversity and struggle to generate high-quality long-tail corner cases, thereby restricting their effectiveness for downstream perception.

**Video Editing in Autonomous Driving.** To enrich scene diversity, object-level editing methods insert new objects into existing videos using reference images and 3D bounding boxes (Singh et al., 2024; Liang et al., 2025b; Buburuzan et al., 2025; An et al., 2025a). More recent NeRF- (Mildenhall et al., 2021) and 3DGS-based (Kerbl et al., 2023) approaches (Chen et al., 2021; Huang et al., 2024b; Chen et al., 2024; Wei et al., 2024; Zhu et al., 2025) improve geometric fidelity, yet remain limited by sparse views, inconsistent lighting, and restricted background diversity. In contrast, our approach builds on generative models and introduces a novel 3D-aware video editing mechanism, enabling seamless insertion of diverse 3D assets and producing geometrically and visually diverse data that effectively boosts downstream performance.

## 3 DREAM4DRIVE

Our goal is to generate high-quality synthetic videos based on real video with ground-truth 3D box annotations and a target 3D asset, creating synthetic data for training downstream perception models. We first introduce the preliminaries in Sec. 3.1, then explain how to conduct 3D-aware scene editing in Sec. 3.2, and finally describe how to render the edited video from the guidance maps in Sec. 3.3. The overall framework is shown in Fig. 2.

### 3.1 PRELIMINARIES

**Latent Diffusion Models** (LDMs) (Rombach et al., 2022) address the high computational cost of diffusion models by operating in a lower-dimensional latent space. Given an image $\mathbf{x}$, an encoder $\mathcal{E}$ is used to obtain the corresponding latent representation $\mathbf{z} = \mathcal{E}(\mathbf{x})$. The forward diffusion process in the latent space is defined as a gradual noising process:

$$q(\mathbf{z}_t|\mathbf{z}_0) = \mathcal{N}(\mathbf{z}_t; \sqrt{\bar{\alpha}_t}\mathbf{z}_0, (1 - \bar{\alpha}_t)\mathbf{I}), \tag{1}$$

where $\bar{\alpha}_t = \prod_{i=1}^{t}(1 - \beta_i)$ with $\beta_i$ being a variance schedule. The reverse process is modeled by a neural network $\epsilon_\theta$ and parameterized as:

$$p_\theta(\mathbf{z}_{t-1}|\mathbf{z}_t) = \mathcal{N}(\mathbf{z}_{t-1}; \mu_\theta(\mathbf{z}_t, t), \Sigma_\theta(\mathbf{z}_t, t)), \tag{2}$$

Finally, a decoder $\mathcal{D}$ maps the denoised latent variable back to the image space, i.e., $\mathbf{x}' = \mathcal{D}(\mathbf{z}_0)$.

**ControlNets** (Zhang et al., 2023a) extend LDMs by incorporating additional conditioning signals $\mathbf{c}$ to provide finer control over the generation process. In particular, the reverse diffusion process is modified to condition on $\mathbf{c}$:

$$p_\theta(\mathbf{z}_{t-1}|\mathbf{z}_t, \mathbf{c}) = \mathcal{N}(\mathbf{z}_{t-1}; \mu_\theta(\mathbf{z}_t, t, \mathbf{c}), \Sigma_\theta(\mathbf{z}_t, t, \mathbf{c})), \tag{3}$$

The conditioning variable $\mathbf{c}$ can incorporate various forms of guidance, including spatial maps, semantic layouts, and other task-specific signals. By integrating $\mathbf{c}$, ControlNets allow for more precise manipulation of the generation process and improving the quality of synthesized images.

### 3.2 3D-AWARE SCENE EDITING

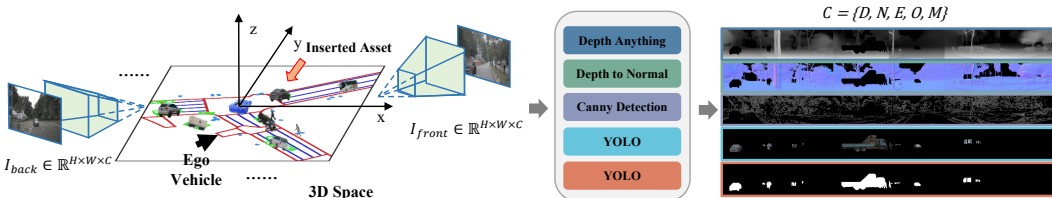

Figure 3: The illustration of 3D-aware scene editing. Given the input images, we first obtain the depth map, normal map, and edge map for the background and then render the object image and object mask for the target 3D asset.

To overcome the limitations of implicit 3D controls (like bounding boxes or BEV maps) in precisely placing assets, we introduce a new form of 3D-aware guidance maps.

For an input RGB image $I \in \mathbb{R}^{H \times W \times 3}$, we use Depth Anything (Yang et al., 2024) to obtain the depth map $D \in \mathbb{R}^{H \times W \times 1}$. The normal map $N \in \mathbb{R}^{H \times W \times 3}$ is then derived from the depth map. We also use OpenCV's Canny edge detector to get the edge map $E \in \mathbb{R}^{H \times W \times 1}$. It is important to note that the depth, normal, and edge information within the foreground object regions are masked since our model needs to learn to generate an edited video based on the target 3D asset.

For a target 3D asset in our DriveObj3D, we position it within the 3D space of the original video based on the provided 3D bounding boxes $\{B_i\}_{i=1}^{T}$. For each frame and each view, we then use calibrated camera intrinsics $K_v$ and extrinsics $E_v$ to render the target 3D asset. Therefore, we can obtain the object image $O \in \mathbb{R}^{H \times W \times 3}$ and object mask $M \in \mathbb{R}^{H \times W \times 1}$.

Once we have obtained the depth map, normal map, and edge map for the background, as well as the rendered object image and object mask for the target 3D asset, we utilize a fine-tuned driving

world model to render the edited video based on the 3D-aware guidance maps. This 3D-aware scene editing pipeline effectively utilizes the accurate pose, geometry, and texture information provided by 3D assets, ensuring geometric consistency in the results generated. Notably, our method does not depend on 3D bounding box embeddings for controlling object placement. Instead, we directly edit in 3D space, offering a more intuitive and reliable way to manage control.

## 3.3 3D-AWARE VIDEO RENDERING

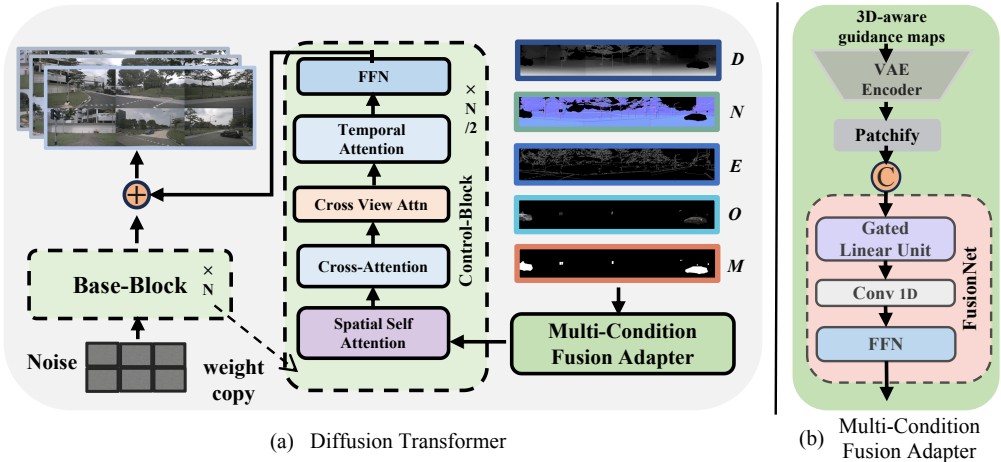

(a) Diffusion Transformer

(b) Multi-Condition Fusion Adapter

Figure 4: The illustration of 3D-aware video rendering. Given the 3D-aware guidance maps, we employ a multi-condition fusion adapter to control the video generation of a diffusion transformer, rendering the edited video.

The video generation process in recent methods (Wen et al., 2024; Li et al., 2024a; Guo et al., 2025) relies on sparse spatial controls, such as bird's eye view (BEV) maps and projected 3D bounding boxes. While this approach allows for some control over the synthesized content, it often falls short in achieving high-quality generation. Instead of relying on sparse spatial controls, we fine-tune a driving world model to generate edited videos from the dense 3D-aware guidance maps, including depth map ($D$), normal map ($N$), and edge map ($E$) for the background and the image ($O$) and mask ($M$) for the foreground object.

The architecture of the multi-condition fusion adapter is shown in Fig. 4. We first encode the five conditions using a VAE, and then apply different 3D embedders to patchify the latents. A FusionNet module then combines these five sets of features, as described by the following equation:

$$F_{\text{fusion}} = \text{FusionNet}\left(\bigoplus_{k=1}^{5} 3\text{DEmbedder}_k(\text{VAE}(C_k)) \ \middle| \ C_k \in \{D, N, E, O, M\}\right), \qquad (4)$$

where $D, N, E, O, M$ denote the depth, normal, edge, object, and mask, respectively, and $\oplus$ indicates concatenation along the channel dimension. The fused features are incorporated into the control blocks of the DiT architecture, enriching semantic information and thereby facilitating instance-level spatial alignment, temporal consistency, and semantic fidelity. The outputs of the control blocks are further integrated into the base block. Additionally, a spatial view attention mechanism is introduced to enhance cross-view coherence, which is especially beneficial in driving scenes.

We adopt rectified flow (Liu et al., 2022) for stable sampling and classifier-free guidance (Ho & Salimans, 2022) to balance text with multiple 3D geometric conditions, thereby improving controllability. The main training objective is a simplified diffusion loss that predicts the noise component at each timestep:

$$\mathcal{L}_{\text{diffusion}} = \mathbb{E}_{t, \mathbf{z}_0, \epsilon \sim \mathcal{N}(0, \mathbf{I})}\left[||\epsilon - \epsilon_\theta(\mathbf{z}_t, t, \mathbf{c})||^2\right], \qquad (5)$$

where $\epsilon$ is Gaussian noise, $\epsilon_\theta$ the predicted noise, $\mathbf{z}_t$ the noisy latent at timestep $t$, and $\mathbf{c}$ all conditioning signals. To achieve precise instance-level control, we introduce a Foreground Mask Loss as Ji et al. (2025) and an LPIPS loss (Zhang et al., 2018). The final training objective is:

$$\mathcal{L}_{\text{total}} = \lambda_{\text{diffusion}}\mathcal{L}_{\text{diffusion}} + \lambda_{\text{mask}}\mathcal{L}_{\text{mask}} + \lambda_{\text{lpips}}\mathcal{L}_{\text{LPIPS}}, \qquad (6)$$

In our experiments, the weights are empirically set as $\lambda_{\text{diffusion}} = 1.0$, $\lambda_{\text{mask}} = 0.1$, and $\lambda_{\text{lpips}} = 0.1$. Notably, our training framework requires no expensive 3D annotations, relying solely on RGB videos and their 3D-aware guidance maps, which can be generated in real time using off-the-shelf tools. This approach significantly reduces training costs. More details are included in the Appx. A.

## 4 DRIVEOBJ3D

The diversity of the synthesized scenes is directly determined by the richness of the asset library used for insertion. To build a large-scale 3D assets for diverse 3D-aware video editing, we design a simple yet effective pipeline. The core idea is to decompose the asset generation process into three steps: (i) 2D instance segmentation; (ii) multi-view image generation; (iii) 3D mesh generation. As shown in Fig. 5, the pipeline inputs a video or image and the target asset category, and generates a 3D mesh for downstream applications.

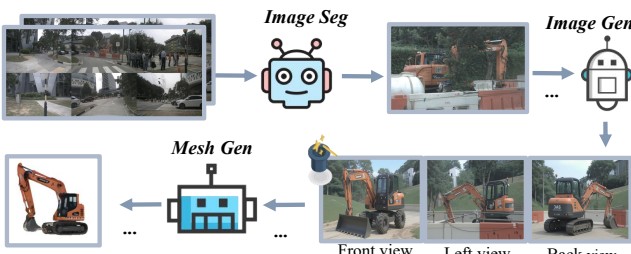

Figure 5: The illustration of creating a 3D asset in DriveObj3D. We first apply a segmentation model to segment the target object, then generate multi-view images, and finally create a 3D mesh from those images.

Concretely, we first localize and segment the target object. Given an input image $I$ and category label *class*, Grounded-SAM (Ren et al., 2024) is applied to segment the object $I_{\text{target}}$. To overcome the occlusion of target object, we employ a multi-view image generation model Qwen-Image (Wu et al., 2025). Conditioned on $I_{\text{target}}$, Qwen-Image synthesizes a set of novel views $\{I_v\}_{v=1}^{N}$, which are subsequently fed into a multi-view reconstruction model Hunyuan3D (Hunyuan3D et al., 2025) to recover the final 3D mesh.

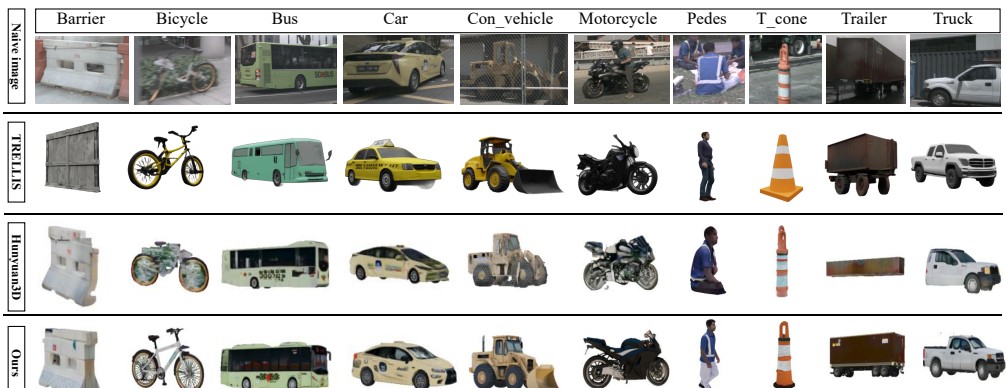

Figure 6: Comparison of 3D asset generation across different methods. Our simple yet effective method produces better 3D assets across diverse categories in autonomous driving, outperforming existing baselines. Con_vehicle is construction vehicle; Pdes is Pedestrian; T_cone is traffic_cone.

As shown in Fig. 6, assets generated by Text-to-3D methods (Xiang et al., 2025) often exhibit style inconsistencies with the original data, while single-view approaches (Hunyuan3D et al., 2025) tend to produce incomplete assets. In contrast, our simple yet effective pipeline leverages multi-view synthesis to generate complete and high-fidelity assets, even under severe occlusions. To support large-scale downstream driving tasks, we construct a diverse asset dataset, DriveObj3D, covering a wide range of categories in driving scenarios in Fig. 7. All assets will be released publicly.

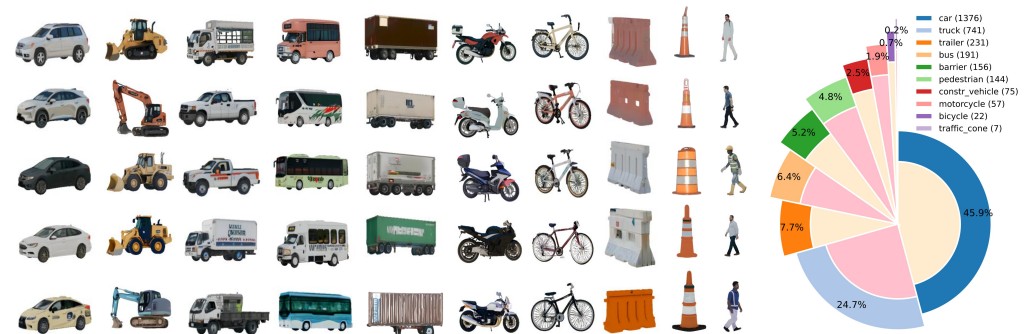

Figure 7: **DriveObj3D** dataset. A large-scale collection of diverse 3D assets across typical driving categories, supporting scalable video editing and synthetic data generation.

| | 1x Epochs | | 2x Epochs | | | | | | |
|---|---|---|---|---|---|---|---|---|---|
| | Real (28130) | Dream4Drive (+420, <2%) | Real (28130) | DriveDreamer (Wang et al., 2024a) | WoVoGen* (Lu et al., 2024) | MagicDrive (Gao et al., 2023) | Panacea (Wen et al., 2024) | SubjectDrive (Huang et al., 2024a) | Dream4Drive (Ours) |
| mAP ↑ | 34.5 | **36.1** | 38.4 | 35.8 | 36.2 | 35.4 | 37.1 | 38.1 | **38.7** |
| mAVE ↓ | 29.1 | **28.9** | 27.7 | – | 123.4 | – | 27.3 | **26.4** | 26.8 |
| NDS ↑ | 46.9 | **47.8** | 50.4 | 39.5 | 18.1 | 39.8 | 49.2 | 50.2 | **50.6** |

Table 1: Comparison of detection under different training epochs. ∗ indicates the evaluation of WoVoGen is only on the vehicle classes of cars, trucks, and buses.

## 5 EXPERIMENTS

### 5.1 SETUP

**Datasets.** We utilize the nuScenes dataset (Caesar et al., 2020) for building 3D assets and finetuning our video generation model. It comprises 1000 scenes in total, with 700 designated for training, 150 for validation, and 150 for testing. Each scene contains a 20-second multi-view video captured by 6 cameras. More details on inserted scene and asset selection are given in the Appx. A.

**Metrics.** Following Panacea (Wen et al., 2024) and SubjectDrive (Huang et al., 2024a), we primarily evaluate how the generated data improves perceptual model performance on detection and tracking tasks. Detection metrics include nuScenes Detection Score (NDS), mean Average Precision (mAP), mean Average Orientation Error (mAOE), and mean Average Velocity Error (mAVE). Tracking metrics include Average MultiObject Tracking Accuracy (AMOTA), Precision (AMOTP), Recall, and Accuracy (MOTA).

### 5.2 MAIN RESULTS

**Effectiveness for Downstream Tasks.** We evaluate the effectiveness of Dream4Drive against prior driving world models, with detection and tracking results reported in Tab. 1 and Tab. 2. While Panacea and SubjectDrive outperform the real dataset baseline at double training epochs, aligning the training epochs shows minimal gains over using real data alone. In contrast, our method explicitly edits objects at specified 3D positions and leverages 3D-aware guidance maps to guide foreground-background synthesis, generating accurately annotated videos that consistently improve downstream perception models. Remarkably, with only 420 inserted samples, our approach outperforms prior methods that used the full set of synthetic data. Moreover, for the first time, synthetic data achieves performance that surpasses real data when training epochs are equal.

**Effectiveness for Various Resolutions.** As generative models continue to advance, the ability to synthesize high-resolution videos has become achievable (Gao et al., 2024b;a). To investigate the effect of high-resolution synthetic data on downstream perception models, we further conduct experiments for detection and tracking tasks at a resolution of $512 \times 768$, as reported in Tab. 3 and 4.

Under both 1x and 2x epochs, real data and Dream4Drive significantly outperform their low-resolution counterparts ($256 \times 512$, Tab. 1 and 2). Remarkably, with high-resolution augmentation, Dream4Drive requires only 420 samples to achieve a 4.6 point (12.7%) mAP increase and a 4.1 point (8.6%) NDS improvement. Most of the gains come from large vehicle categories, including bus, construction_vehicle, and truck; detailed AP for each category is provided in the Appx. B.

| | 1x Epochs | | 2x Epochs | | | |
|---|---|---|---|---|---|---|
| | Real (28130) | Dream4Drive (+420, <2%) | Real (28130) | Panacea (Wen et al., 2024) | SubjectDrive (Huang et al., 2024a) | Dream4Drive (Ours) |
| AMOTA ↑ | 30.1 | **31.2** | 34.1 | 33.7 | 33.7 | **34.4** |
| AMOTP ↓ | 137.9 | **135.4** | 134.1 | 135.3 | 135.3 | **133.5** |

Table 2: Comparison of tracking under different training epochs.

| | 1x Epochs | | | 2x Epochs | | | 3x Epochs | | |
|---|---|---|---|---|---|---|---|---|---|
| | Real | Naive Insert | Dream4Drive | Real | Naive Insert | Dream4Drive | Real | Naive Insert | Dream4Drive |
| mAP ↑ | 36.1 | 40.1 | **40.7** | 42.2 | 42.9 | **43.6** | 43.1 | 43.1 | **44.5** |
| mATE ↓ | 69.2 | 64.7 | **64.2** | 61.6 | 62.4 | **61.6** | 60.5 | 61.5 | **59.8** |
| mAOE ↓ | 56.7 | 49.0 | **48.0** | 43.2 | **37.5** | 39.4 | 45.7 | **38.9** | 40.1 |
| mAVE ↓ | 28.5 | 28.4 | **27.1** | 27.5 | **27.3** | 27.4 | 27.4 | 27.4 | **27.2** |
| NDS ↑ | 47.9 | 51.3 | **52.0** | 53.2 | 54.0 | **54.3** | 53.6 | 54.2 | **55.0** |

Table 3: Detection performance under different training epochs (1x, 2x, 3x). "Naive Insert" denotes the direct projection of 3D assets into the original scene. Results are reported at 512×768 resolution.

Unlike prior augmentation paradigms, Dream4Drive consistently outperforms training on real data alone, regardless of the number of epochs, highlighting the value of high-quality synthetic data.

**Quantitative and Qualitative Comparison with Naive Insertion.**   After extracting 3D assets, we can directly generate edited videos with projection. To assess the impact of 3D-aware video rendering versus direct insertion on downstream tasks, we conduct comprehensive evaluations, as reported in Tab. 3 and 4. While direct insertion improves performance over real data alone, its results remain inferior to our generative method due to missing realism, such as shadows and reflections. Interestingly, direct insertion achieves the highest mAOE, likely because the inserted assets perfectly align with the original bounding box orientations. Fig. 8 presents visual comparisons between naive insertion and our generative approach across multiple scenes.

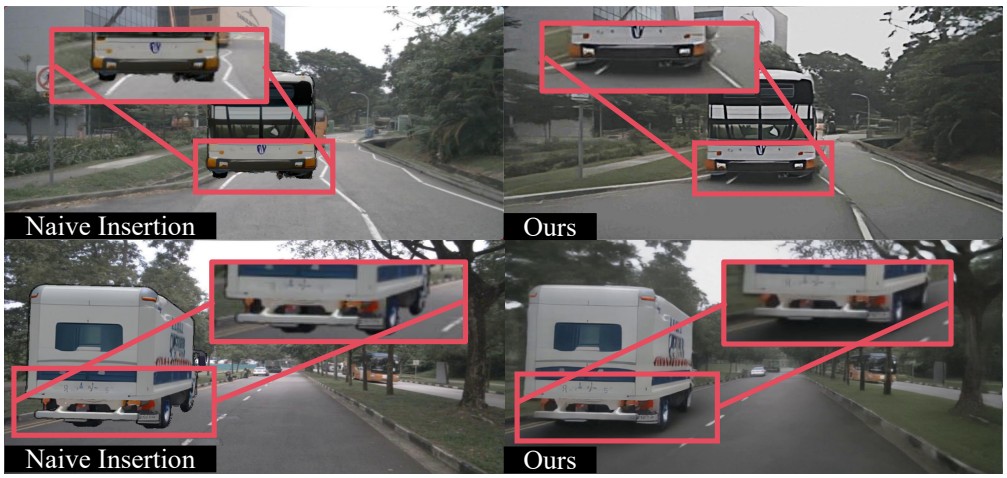

Figure 8: Comparison with naive insertion. As can be seen, Dream4Drive generates more realistic edited videos than naive insertion.

**Comprehensive visualization of asset insertion results.**   More asset categories, insertion locations, and corner case scenarios are comprehensively presented in the Appx. D.

## 5.3   ABLATION STUDIES

**Effect of Insertion Position.**   Dream4Drive can edit videos by projecting 3D assets anywhere in a scene. To systematically evaluate the impact of insertion position on downstream model performance, we categorize positions as front, back, left, and right, as shown in Tab. 5.

|  | 1x Epochs | | | 2x Epochs | | | 3x Epochs | | |
|---|---|---|---|---|---|---|---|---|---|
|  | Real | Naive Insert | Ours | Real | Naive Insert | Ours | Real | Naive Insert | Ours |
| AMOTA ↑ | 32.8 | 36.5 | **37.9** | 39.7 | 42.2 | **42.6** | 41.3 | 42.2 | **43.5** |
| AMOTP ↓ | 134.0 | 128.7 | **128.0** | 125.1 | 124.0 | **123.3** | 124.1 | 123.7 | **121.3** |
| MOTA ↑ | 28.1 | 31.7 | **33.1** | 35.6 | 37.3 | **37.4** | 36.8 | 37.5 | **38.5** |
| RECALL ↑ | 44.0 | 45.4 | **46.9** | 50.7 | 51.1 | **51.8** | 52.4 | 51.8 | **52.5** |

Table 4: Tracking performance under different training epochs (1x, 2x, 3x). "Naive Insert" denotes the direct projection of 3D assets into the original scene. Results are reported at 512×768 resolution.

|  | Views | | | | Distances | | | 3D Asset Generation Methods | | | Speed | | |
|---|---|---|---|---|---|---|---|---|---|---|---|---|---|
|  | Front | Back | Left | Right | Close | Mid | Far | Trellis | Hunyuan3D | Ours | Low | Middle | High |
| mAP ↑ | 40.2 | 40.2 | 40.2 | 39.8 | 39.7 | 40.3 | **40.5** | 39.8 | 40.2 | **40.7** | 40.4 | 40.7 | **40.9** |
| mATE ↓ | 66.2 | 66.0 | **64.6** | 66.2 | 65.7 | 65.4 | 65.1 | 65.6 | 65.1 | **64.2** | 64.5 | 64.2 | **63.9** |
| mAOE ↓ | 51.2 | 55.1 | **45.7** | 51.4 | 52.2 | 51.7 | 49.7 | 51.8 | 50.8 | **48.0** | 48.3 | 48.1 | 48.0 |
| mAVE ↓ | 27.7 | **27.5** | 28.5 | 27.9 | 28.1 | 28.0 | 27.9 | 27.8 | 28.0 | **27.1** | 27.1 | 27.1 | **27.0** |
| NDS ↑ | 51.0 | 50.6 | **51.6** | 50.7 | 50.5 | 50.9 | **51.3** | 50.8 | 50.9 | **52.0** | 51.8 | 52.0 | **52.1** |

Table 5: Ablation Studies. We report detection performance across insertion positions and asset, with best results per block (Views, Distances, 3D Methods) in bold, at 512×768 resolution.

Results indicate that inserting assets in the front or back yields similar performance, whereas left-side insertions outperform right-side ones, with a 0.4 point increase in mAP, 0.9 point increase in NDS, and a 5.7 point reduction in mAOE. This likely indicates dataset bias: most vehicles appear on the ego vehicle's left side, so enhancing such corner cases improves model predictions, while right-side corner cases yield limited gains on the validation set.

We further examine the effect of insertion distance on performance in Tab. 5. Close insertions tend to perform poorly, likely due to the asset blocking the camera view, which interferes with the training of other instances. Distant insertions offer more effective augmentation, as detectors often struggle with distant objects. Increasing their prevalence enhances detection performance.

**Effect of 3D Asset Source.** We observe that the source of inserted assets affects the quality of synthetic data, consistent with (Ljungbergh et al., 2025). We therefore investigate how different asset sources influence downstream perception performance.

As shown in Tab. 5, although Trellis (Xiang et al., 2025) can generate high-quality assets, its style does not fully match autonomous driving scenarios, which can lead to artifacts and degraded quality when assets are inserted, negatively affecting downstream tasks. Hunyuan3D (Hunyuan3D et al., 2025)'s single-view generation also underperforms our multiview approach, since single-image assets may be incomplete, whereas our method produces complete, high-quality 3D assets.

## 5.4 TAKEAWAYS

We summarize the main observations from our experiments with perception model augmentation as follows:

- Duplying original layouts to create synthetic data does not improve performance; instead, enhancing scenes by inserting new 3D assets is an effective strategy for augmentation.

- High-resolution synthetic data offers greater benefits for data augmentation.

- The placement of inserted assets influences the effectiveness of augmentation, highlighting biases present in the dataset.

- Insertions at farther distances generally improve performance, while close-range insertions may introduce strong occlusions that hinder training.

- Using assets from the same dataset reduces the domain gap between synthetic and real data, benefiting downstream model training.

## 6 CONCLUSION

In this paper, we find that previous driving world models inaccurately assess the effectiveness of synthetic data for downstream tasks. To address this, we present Dream4Drive, a 3D-aware synthetic data generation pipeline that synthesizes high-quality multi-view corner cases. To facilitate future research, we also contribute a large-scale 3D asset dataset named DriveObj3D, covering the typical categories in driving scenarios. Extensive experiments demonstrate that with less than 2% additional synthetic data, Dream4Drive consistently improves downstream perception, validating the effectiveness of synthetic data for autonomous driving.

## ACKNOWLEDGMENTS

This work is supported by National Natural Science Foundation of China (92470121, 62402016), National Key R&D Program of China (2024YFA1014003), Zhongguancun Academy, (Grant No.s C20250204, C20250602), and High-performance Computing Platform of Peking University.

## ETHICS STATEMENT

This work focuses on synthetic data generation for autonomous driving, aiming to improve downstream perception through the proposed Dream4Drive framework. We build upon publicly available driving datasets and a curated 3D asset library, without collecting new human-subject data or personally identifiable information. Potential risks include the misuse of synthetic driving scenarios for surveillance or malicious applications, as well as biases introduced by the underlying datasets and assets. We encourage responsible use of this research strictly for advancing safe and reliable autonomous driving.

## REPRODUCIBILITY STATEMENT

We ensure reproducibility by providing detailed descriptions of algorithms, metrics, and experimental settings in the main paper and appendix. All assets and code will be released upon publication.

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

# A  IMPLEMENTATION DETAILS

## A.1  MODEL AND TRAINING DETAILS

**DriveObj3D Generation Pipeline.** We adopt Grounded-SAM for image segmentation, Qwen-Image-Edit for image generation, and Hunyuan3D 2.0 for 3D asset generation.

**Inpainting Diffusion Model.** For video synthesis, Dream4Drive is constructed upon a DiT-based architecture. The backbone is initialized with pretrained weights from MagicDriveDiT (Gao et al., 2024a), which originally conditioned on BEV maps and 3D bounding boxes. We extend its ControlNet branch by incorporating novel *3D-aware guidance maps*, and fine-tune the model on the nuScenes dataset. The generated videos have a resolution of $512 \times 768$ with 33 frames.

Training is conducted in `PyTorch` using 8 NVIDIA H200 GPUs with mixed-precision acceleration for 2000 iterations. We employ the AdamW optimizer with a weight decay of 0.01 and adopt a cosine annealing learning rate schedule with linear warm-up over the first 10% of steps. The learning rate is set to $2 \times 10^{-4}$, and the batch size per GPU is 1. During training, we introduce additional Mask Loss and LPIPS Loss to enhance perceptual consistency and local reconstruction quality.

**Mask Loss.** In the VAE-decoded space, we compute the mean squared error (MSE) between prediction and ground truth only within the foreground mask region. Let $\hat{\mathbf{x}}$ and $\mathbf{x}$ denote the predicted and ground-truth decoded images, and $\mathbf{M}$ be a binary mask (1 for constrained pixels, 0 otherwise). The masked MSE is defined as:

$$\mathcal{L}_{\text{mask}} = \frac{|\mathbf{M} \odot (\hat{\mathbf{x}} - \mathbf{x})|_2^2}{\sum \mathbf{M} + \epsilon}, \tag{7}$$

where $\odot$ denotes element-wise multiplication, and the denominator normalizes over valid pixels.

**LPIPS Loss.** To further improve perceptual quality, we adopt the LPIPS (Learned Perceptual Image Patch Similarity) loss (Zhang et al., 2018), which measures feature-level perceptual differences between prediction and ground truth:

$$\mathcal{L}_{\text{LPIPS}} = \text{LPIPS}(\hat{\mathbf{x}}, \mathbf{x}), \tag{8}$$

where the LPIPS network is pretrained on large-scale perceptual similarity data.

## A.2  DETAILED SYNTHESIS PROCEDURE AND TIME-COST ANALYSIS FOR GENERATING THE 420 SAMPLES

We begin by clarifying the construction of the 420 samples, then present a detailed, step-by-step account of the time required, and conclude by describing the operations carried out at each stage.

**Construction of the 420 samples.** We randomly select seven annotated scenes and insert ten object categories (one asset per category). Each video contains 33 frames, from which we extract six key frames for downstream training, resulting in a total of $7 \times 10 \times 6 = 420$ samples.

| Step | Automation (s) | Manual effort (s) |
|---|---|---|
| Scene selection & initial position annotation (MLLM-assisted) | $700 \times 10$ | |
| Scene verification | | $7 \times 20$ |
| Scene orientation annotation | | $7 \times 15$ |
| Asset selection | | $10 \times 5$ |
| Video generation (DiT) | $70 \times 10 \times 4$ | |
| **Total** | $9800s \approx 2.72h$ | 295s |

Table 6: Automation and manual efforts involved in synthesizing the 420 samples.

**Detailed synthesis steps and time analysis (Tab. 6.)**

- **Scene selection and initial position annotation.** An MLLM identifies scenes containing at least one collision-free straight-driving trajectory (front, rear, left, or right). For eligible scenes, it annotates the asset's initial insertion position in ego-vehicle coordinates $(x, y, z)$ for the first frame. The nuScenes training set contains 700 videos, and each MLLM query takes about 10 seconds, resulting in a total of $700 \times 10$ seconds. The full prompt is provided in Listing 1

- **Human verification of MLLM-filtered scenes.** Only seven scenes are needed. We manually verify the MLLM-generated initial positions, requiring approximately 20 seconds per scene, totaling $7 \times 20$ seconds.

- **Scene orientation annotation.** Orientation is calibrated using the `pyrender` package. All assets are pre-standardized to ensure consistent orientation. For each scene, we render an initial asset insertion using nuScenes intrinsics and extrinsics to determine the correct yaw angle. The first rendering takes 6 seconds, the correction takes 3 seconds, and the second rendering takes another 6 seconds, totaling roughly 15 seconds per scene (i.e., $7 \times 15$ seconds).

- **Asset selection.** We require only one asset per object category. A random choice per category suffices (about 5 seconds each), giving a total of $10 \times 5$ seconds.

- **Video generation.** We generate 70 videos for the 420 samples. Each video uses 10 inference steps; on an H200 GPU each step takes roughly 4 seconds, producing a total cost of $70 \times 10 \times 4$ seconds.

**Summary.** We adopt a partially automated pipeline. The full synthesis process for all 420 samples takes under 3 hours, with **approximately 300 seconds of actual human labor**, which is effectively negligible.

## A.3 MORE EXPERIMENT RESULTS

**Quantitative Evaluation of Asset Quality.** We evaluate synthesized 3D assets by measuring image similarity to their corresponding ground-truth originals using CLIP and DINO metrics (Fig. 7). For CLIP model, we adopt `clip-vit-large-patch14-336` Radford et al. (2021), and for DINO model, we use `dinov2-base` Oquab et al. (2023). The results in Tab. 7 show that our pipeline consistently outperforms Trellis and Hunyuan3D across all object categories.

| | | car | truck | bus | trailer | construction vehicle | pedestrian | motorcycle | bicycle | traffic cone | barrier |
|---|---|---|---|---|---|---|---|---|---|---|---|
| | Trellis | 0.8913 | 0.9160 | 0.8930 | 0.9041 | 0.8870 | 0.8753 | 0.9045 | 0.9273 | 0.9254 | 0.9270 |
| CLIP IS | Hunyuan3D | 0.8929 | 0.9043 | 0.9031 | 0.9022 | 0.8787 | 0.8863 | 0.9121 | 0.9170 | 0.9288 | 0.9101 |
| | Ours | **0.9048** | **0.9291** | **0.9197** | **0.9270** | **0.8996** | **0.9061** | **0.9201** | **0.9351** | **0.9305** | **0.9310** |
| | Trellis | 0.7008 | 0.7266 | 0.7012 | 0.6974 | 0.6852 | 0.6721 | 0.6990 | 0.7053 | 0.7275 | 0.7304 |
| DINO IS | Hunyuan3D | 0.7360 | 0.7631 | 0.7365 | 0.7344 | 0.7204 | 0.7074 | 0.7369 | 0.7405 | 0.7610 | 0.7652 |
| | Ours | **0.8117** | **0.7979** | **0.8082** | **0.8040** | **0.7960** | **0.7835** | **0.8065** | **0.8129** | **0.8365** | **0.8394** |

Table 7: Quantitative evaluation of synthesized asset quality using CLIP Image Similarity (CLIP IS) and DINO Image Similarity (DINO IS). Higher scores indicate closer resemblance to the ground-truth assets.

These results align with the downstream perception improvements demonstrated earlier in Tab. 5, confirming the value of high-fidelity asset synthesis.

**Scaling Behavior of OOD Samples.** We further study how downstream performance changes when increasing the amount of OOD synthetic data. As shown in Tab. 8, expanding OOD samples from 7 to 35 scenes does not yield proportional improvements. In fact, too many OOD samples slightly degrade performance, likely because excessive OOD content dilutes in-distribution signal. Small amounts of OOD data improve robustness (Tab. 3), but aggressive scaling is not beneficial.

| | $1\times$ Epochs | | | $2\times$ Epochs | | | $3\times$ Epochs | | |
|---|---|---|---|---|---|---|---|---|---|
| | 7 Scenes | 14 Scenes | 35 Scenes | 7 Scenes | 14 Scenes | 35 Scenes | 7 Scenes | 14 Scenes | 35 Scenes |
| mAP ↑ | 40.7 | 40.4 | 39.7 | 43.6 | 43.1 | 42.3 | 44.5 | 44.1 | 43.6 |
| mATE ↓ | 64.2 | 64.4 | 64.5 | 61.6 | 62.0 | 62.5 | 59.8 | 59.5 | 60.3 |
| mAOE ↓ | 48.0 | 49.7 | 51.6 | 39.4 | 39.8 | 40.2 | 40.1 | 40.3 | 40.4 |
| mAVE ↓ | 27.1 | 27.8 | 28.4 | 27.4 | 28.1 | 28.4 | 27.2 | 27.7 | 27.6 |
| NDS ↑ | 52.0 | 51.6 | 50.9 | 54.3 | 53.8 | 53.1 | 55.0 | 54.6 | 54.2 |

Table 8: Scaling analysis with increasing numbers of OOD scenes under different training epochs. More OOD samples do not necessarily yield better performance.

To examine whether more diverse OOD conditions (rather than more scenes) provide additional gains, we apply style transfer to generate new synthetic data (e.g., rain or nighttime) and augment the original set. Tab. 9 shows consistent improvement across all metrics.

| | 1× Epochs | | 2× Epochs | | 3× Epochs | |
|---|---|---|---|---|---|---|
| | Original (420) | Style Transfer (420+420) | Original (420) | Style Transfer (420+420) | Original (420) | Style Transfer (420+420) |
| mAP ↑ | 40.7 | 41.2 | 43.6 | 44.2 | 44.5 | 44.8 |
| mATE ↓ | 64.2 | 63.7 | 61.6 | 61.2 | 59.8 | 59.7 |
| mAOE ↓ | 48.0 | 47.5 | 39.4 | 39.0 | 40.1 | 39.8 |
| mAVE ↓ | 27.1 | 26.8 | 27.4 | 26.8 | 27.2 | 26.9 |
| NDS ↑ | 52.0 | 52.4 | 54.3 | 54.7 | 55.0 | 55.2 |

Table 9: Effect of style-transferred OOD synthetic data. Adding environmental diversity improves all downstream perception metrics.

**Conclusion.** Our findings indicate that downstream perception benefits most from OOD *scene layouts* and from *environmental diversity*, rather than merely increasing the volume of synthetic data. Additional visualization examples of style transfer are included in the Fig. 20, 19, 21, 22.

### A.4 MORE ABLATION STUDIES

We conduct additional ablation studies on trajectory speed, asset categories, and scene selection diversity. Unless otherwise specified, all experiments are conducted under the 1× training epoch and 512×768 resolution setting.

**Ablation on Trajectory Speed.** Inserted vehicles always follow straight trajectories, and we vary speed across three levels (2 / 5 / 8 m/s). Higher speeds consistently yield larger downstream gains (Tab. 5), consistent with Table 5 in the main paper. Faster inserted vehicles leave the ego-vehicle's vicinity earlier, helping the perception model learn to detect distant objects more effectively.

**Ablation on Asset Categories.** For each object category, we randomly select three assets and split them into three groups (A / B / C). Performance is nearly identical across groups (Tab. 10), demonstrating that visual style differences across assets have negligible influence as long as the OOD layout is preserved.

| | car | truck | bus | trailer | construction vehicle | pedestrian | motorcycle | bicycle | traffic cone | barrier |
|---|---|---|---|---|---|---|---|---|---|---|
| Assets group A | 0.600 | 0.354 | 0.341 | 0.106 | 0.135 | 0.468 | 0.402 | 0.411 | 0.626 | 0.565 |
| Assets group B | 0.594 | 0.350 | 0.343 | 0.110 | 0.135 | 0.471 | 0.405 | 0.414 | 0.625 | 0.563 |
| Assets group C | 0.601 | 0.352 | 0.341 | 0.108 | 0.133 | 0.470 | 0.401 | 0.411 | 0.626 | 0.565 |

Table 10: Ablation on selected asset categories, reporting AP metrics.

**Ablation on Scene Selection Diversity.** We randomly sample two additional sets of insertion scenes, forming three groups (A / B / C) including the original. Downstream results remain similar across groups (Tab. 11), indicating that scene-selection-induced bias is minimal.

| | 1× Epochs | | | 2× Epochs | | | 3× Epochs | | |
|---|---|---|---|---|---|---|---|---|---|
| | A | B | C | A | B | C | A | B | C |
| mAP ↑ | 40.7 | 40.8 | 40.7 | 43.6 | 43.5 | 43.6 | 44.5 | 44.3 | 44.5 |
| mATE ↓ | 64.2 | 64.1 | 64.2 | 61.6 | 61.5 | 61.4 | 59.8 | 59.8 | 59.9 |
| mAOE ↓ | 48.0 | 48.0 | 48.1 | 39.4 | 39.4 | 39.5 | 40.1 | 40.2 | 40.1 |
| mAVE ↓ | 27.1 | 26.9 | 27.1 | 27.4 | 27.5 | 27.3 | 27.2 | 27.3 | 27.2 |
| NDS ↑ | 52.0 | 52.1 | 52.1 | 54.3 | 54.2 | 54.3 | 55.0 | 55.0 | 55.1 |

Table 11: Ablation on insertion-scene selection.

## B DETAILED AP METRICS

Tab. 3 compares detection metrics across different training epochs. Detailed per-class AP scores are provided in Tab. 12, 13, and 14.

## C ADDITIONAL EXPERIMENTS ON GENERATION QUALITY

| Method | car | truck | bus | trailer | construction_vehicle | pedestrian | motorcycle | bicycle | traffic_cone | barrier |
|---|---|---|---|---|---|---|---|---|---|---|
| Real | 0.562 | 0.323 | 0.296 | 0.067 | 0.111 | 0.422 | 0.377 | 0.371 | 0.569 | 0.515 |
| Ours | 0.600 | 0.354 | 0.341 | 0.106 | 0.135 | 0.468 | 0.402 | 0.411 | 0.626 | 0.565 |

Table 12: AP comparison across different categories for Real and Ours (+420) at 1× training epoch.

| Method | car | truck | bus | trailer | construction_vehicle | pedestrian | motorcycle | bicycle | traffic_cone | barrier |
|---|---|---|---|---|---|---|---|---|---|---|
| Real | 0.622 | 0.371 | 0.375 | 0.154 | 0.132 | 0.493 | 0.430 | 0.400 | 0.651 | 0.589 |
| Ours | 0.633 | 0.383 | 0.389 | 0.168 | 0.147 | 0.497 | 0.446 | 0.434 | 0.647 | 0.604 |

Table 13: AP comparison across different categories for Real and Ours (+420) at 2× training epoch.

**Controllability.** The controllability of our approach is quantitatively evaluated using perception metrics from StreamPETR Wang et al. (2023a). We first generate the entire nuScenes validation set with Dream4Drive, and then measure perception performance using a pre-trained StreamPETR model. The relative metrics, compared to those obtained on real data, indicate how well the generated samples align with the original annotations. As shown in Tab. 16, Dream4Drive achieves a relative performance of 82%, demonstrating precise control over foreground object locations.

| Method | FVD↓ | FID↓ |
|---|---|---|
| No cutout&mask | 66.36 | 14.38 |
| No depth&normal | 34.64 | 7.33 |
| No edge | 49.45 | 8.44 |

Table 15: Ablation study on 3D-aware guidance maps.

**Generation Quality.** As shown in Tab. 17, our method achieves FVD 31.84 and FID 5.80, outperforming layout- and 3D semantics-guided methods Gao et al. (2023; 2024a); Li et al. (2024a); Ji et al. (2025). The results indicate improved motion consistency and preserved visual fidelity, confirming that our 3D-aware guidance maps effectively generate both foreground and background content. Notably, **no post-processing or selective filtering** was applied to the generated videos.

**Additional Ablation Studies.** We perform ablation experiments on the components of the Multi-condition Fusion Adapter, with results presented in Tab. 15. The findings indicate that all components contribute significantly to overall performance.

# D ADDITIONAL VISUALIZATION

We present more comparisons between the naive insert and ours in Fig. 9, 10, asset insertion videos across different categories in Fig. 11, 12, 13, 14, 15, and 16, where all inserted assets are highlighted with red bounding boxes. In addition, we showcase several corner-case examples, such as imminent collision and close-following scenarios, also illustrated in Fig. 17 and Fig. 18.

# E LIMITATION AND FUTURE WORKS

Although Dream4Drive is capable of inserting arbitrary assets into diverse scenes, automatically ensuring that the inserted trajectories remain within drivable areas and avoid collisions with pedestrians or other vehicles remains an open challenge. Addressing this issue would enable more flexible generation of diverse corner cases.

# F STATEMENT ON LLM USAGE

During the preparation of this manuscript, large language models (LLMs) were used solely for language refinement. All scientific ideas, experiments, analyses, and conclusions are the authors' original contributions.

| Method | car | truck | bus | trailer | construction_vehicle | pedestrian | motorcycle | bicycle | traffic_cone | barrier |
|---|---|---|---|---|---|---|---|---|---|---|
| Real | 0.627 | 0.362 | 0.367 | 0.154 | 0.132 | 0.488 | 0.436 | 0.454 | 0.656 | 0.632 |
| Ours | 0.639 | 0.377 | 0.408 | 0.190 | 0.164 | 0.501 | 0.446 | 0.439 | 0.654 | 0.621 |

Table 14: AP comparison across different categories for Real and Ours (+420) at 3× training epoch.

| Method | mAP ↑ | mATE ↓ | mAOE ↓ | mAVE ↓ | NDS ↑ |
|---|---|---|---|---|---|
| Real | 36.1 | 69.2 | 56.7 | 28.5 | 47.9 |
| Gen-nuScenes | 24.4 | 78.5 | 66.1 | 34.9 | 39.1 (82%) |

Table 16: Domain gap. Comparison of detection performance on Real and Gen-nuScenes validation sets at 512×768 resolution. Values are reported in percentage.

| Method | FPS | Resolution | FVD↓ | FID↓ |
|---|---|---|---|---|
| MagicDrive Gao et al. (2023) | 12Hz | 224×400 | 218.12 | 16.20 |
| Panacea Wen et al. (2024) | 2Hz | 256×512 | 139.00 | 16.96 |
| SubjectDrive Huang et al. (2024a) | 2Hz | 256×512 | 124.00 | 15.98 |
| DriveWM Wang et al. (2024b) | 2Hz | 192×384 | 122.70 | 15.80 |
| Delphi Ma et al. (2024a) | 2Hz | 512×512 | 113.50 | 15.08 |
| MagicDriveDiT Gao et al. (2024a) | 12Hz | 224×400 | 94.84 | 20.91 |
| DiVE Jiang et al. (2024) | 12Hz | 480×854 | 94.60 | - |
| UniScene Li et al. (2024a) | 12Hz | 256×512 | 70.52 | 6.12 |
| CoGen Ji et al. (2025) | 12Hz | 360×640 | 68.43 | 10.15 |
| DriveDream-2 Zhao et al. (2025a) | 12Hz | 512×512 | 55.70 | 11.20 |
| Ours | 12Hz | 512×768 | **31.84** | **5.80** |

Table 17: Quantitative comparison on video generation quality with other methods. Our method achieves the best FVD and FID score.

```
You are an expert in autonomous-driving scene understanding.
Based on the input video, perform the following tasks without any bias or preference toward
    specific scene types (e.g., do not favor simple scenes over complex ones, and do not
    assume safety based on convenience). All decisions must be strictly derived from
    observable evidence in the video.

[Task Objective]
Evaluate the driving scene and determine whether there exists at least one safe, unobstructed,
     straight driving trajectory in any of the four directions relative to the ego vehicle:
    front, rear, left, or right.

If such a direction exists, identify the direction and provide the recommended initial
    insertion position for a new 3D asset. The position must be given in the ego-vehicle
    coordinate system and must correspond to the first frame of the video.

[Ego-Vehicle Coordinate System]
- The ego vehicle is located at the origin (0, 0, 0)
- x-axis: forward
- y-axis: left
- z-axis: upward

[Definition of "Safe Straight Trajectory"]
A direction is considered valid for 3D-asset insertion only if all the following conditions
    are met:
1. Throughout the entire video, that direction remains free of vehicles, pedestrians, and any
    static or dynamic obstacles blocking straight motion.
2. The road in that direction is continuous and drivable (not leading into oncoming traffic
    lanes, sidewalks, barriers, buildings, etc.).
3. There is sufficient spatial clearance for inserting a typical vehicle-sized 3D asset (
    approx. length 4m, width 1.8m, height 1.5m).
4. The insertion location must not collide with any existing object in the first frame.
5. No bias is allowed - judge each direction solely based on objective video evidence, without
     assuming simplicity or difficulty, and without preferring any type of scene.
```

```
[Output Format]
Respond strictly in the following JSON structure:

{
 "is_insertable": true/false,
 "direction": "front" | "rear" | "left" | "right" | null,
 "initial_position": {
    "x": number or null,
    "y": number or null,
    "z": number or null
 },
 "reason": "Brief explanation of the decision"
}

[Examples]

If the front direction is clear and drivable:
{
 "is_insertable": true,
 "direction": "front",
 "initial_position": {"x": 8.0, "y": 0.0, "z": 0.0},
 "reason": "The front lane is unobstructed and drivable for a long distance."
}

If no direction is suitable:
{
 "is_insertable": false,
 "direction": null,
 "initial_position": {"x": null, "y": null, "z": null},
 "reason": "All directions contain obstacles that block a safe straight trajectory."
}
```

Listing 1: Prompt for instructing MLLM to identify safe driving scenarios.

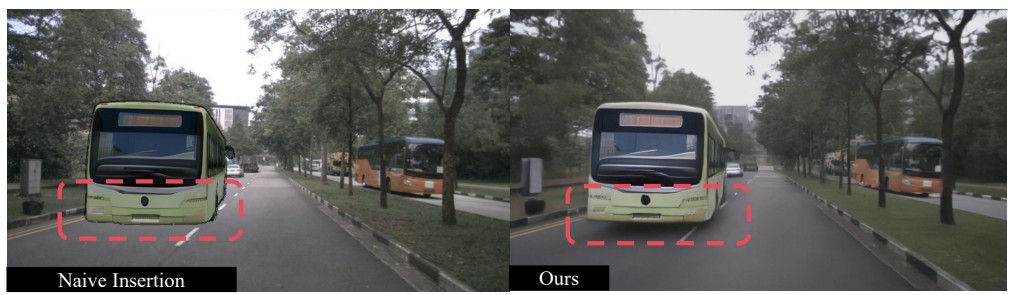

Figure 9: A case study of contrast between reflection and shadow.

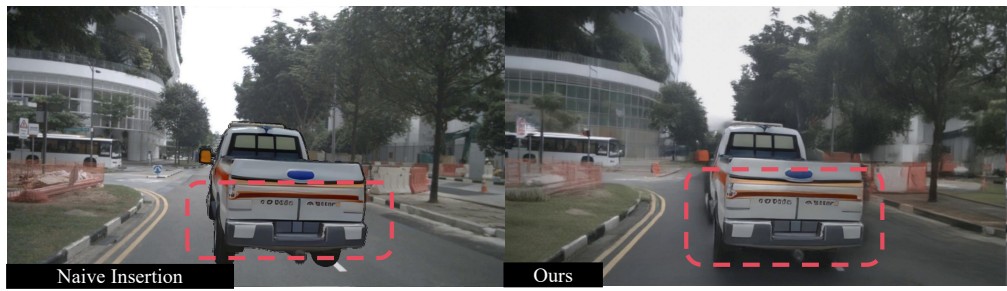

Figure 10: A case study of contrast between reflection and shadow.

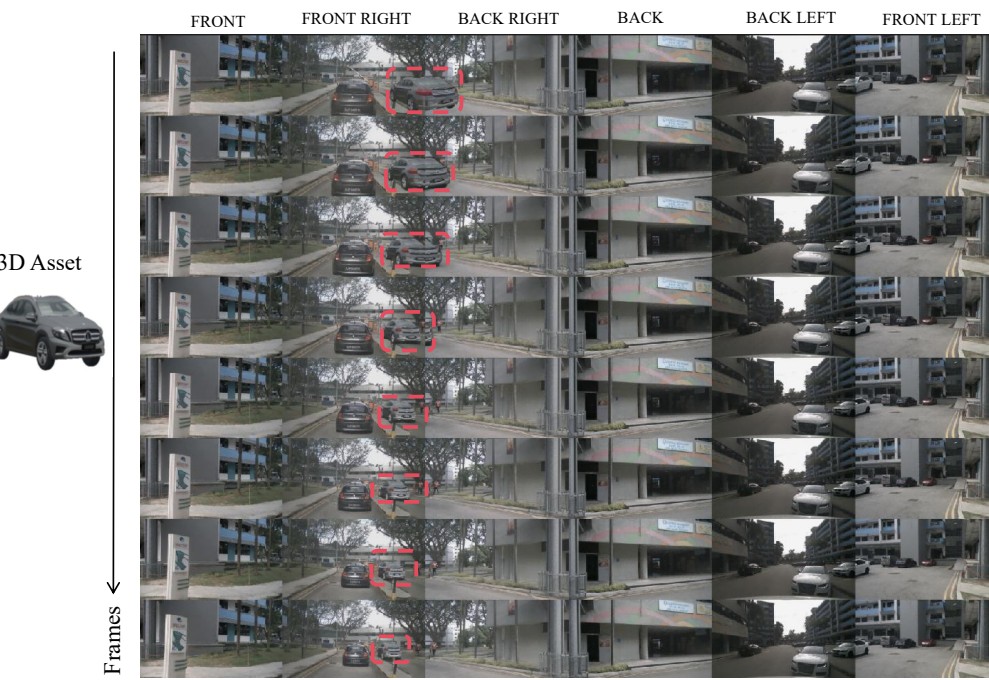

Figure 11: Insertion of a car in the right-side region.

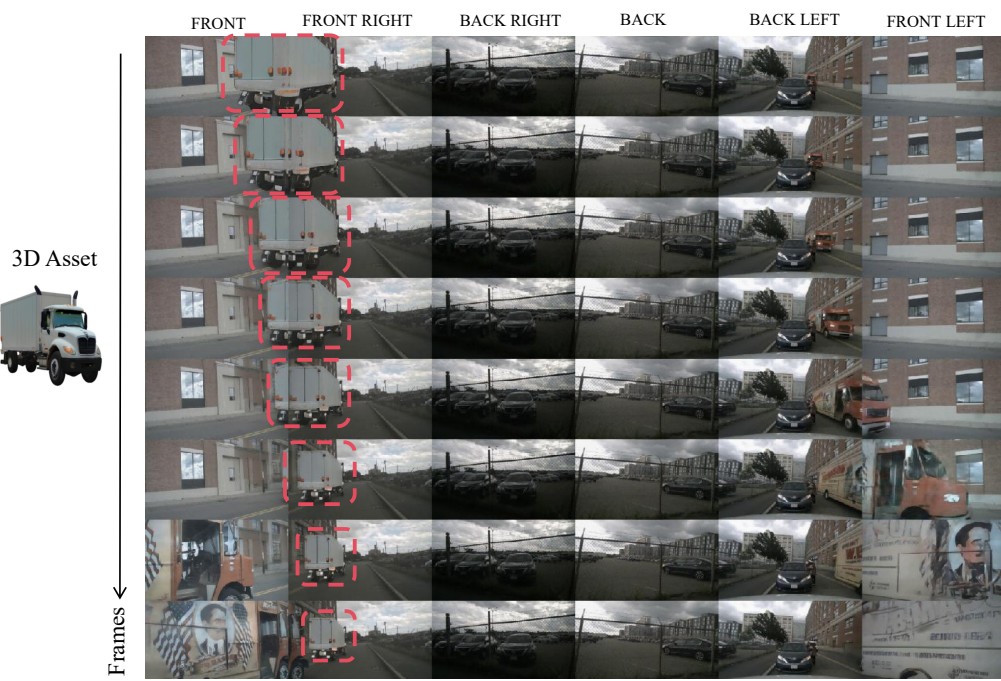

Figure 12: Insertion of a truck in the left-side region.

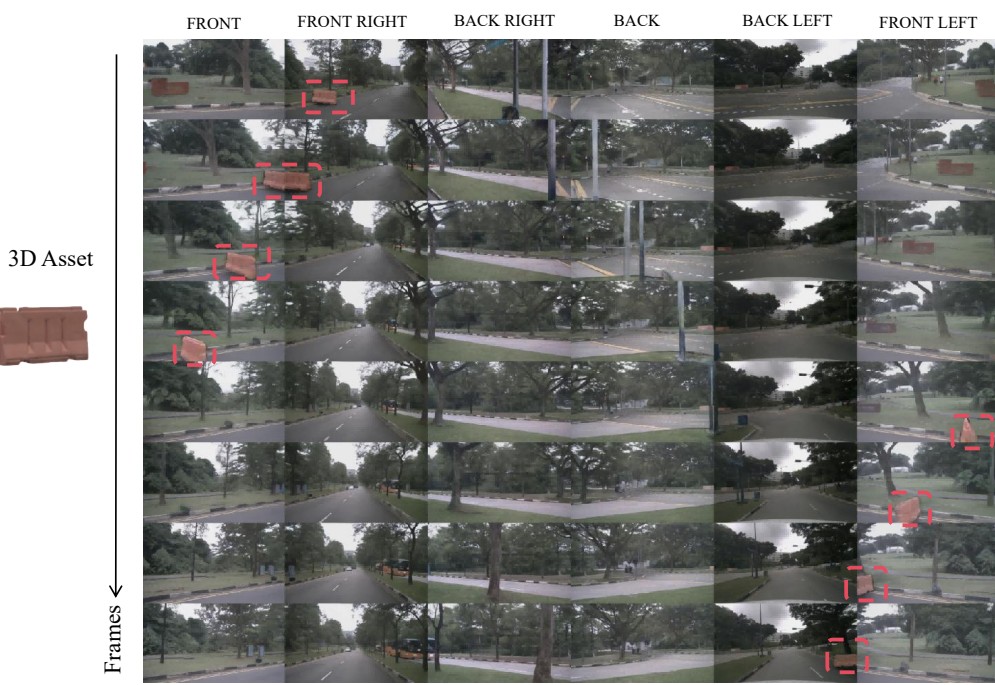

Figure 13: Insertion of a barrier in the left-side region.

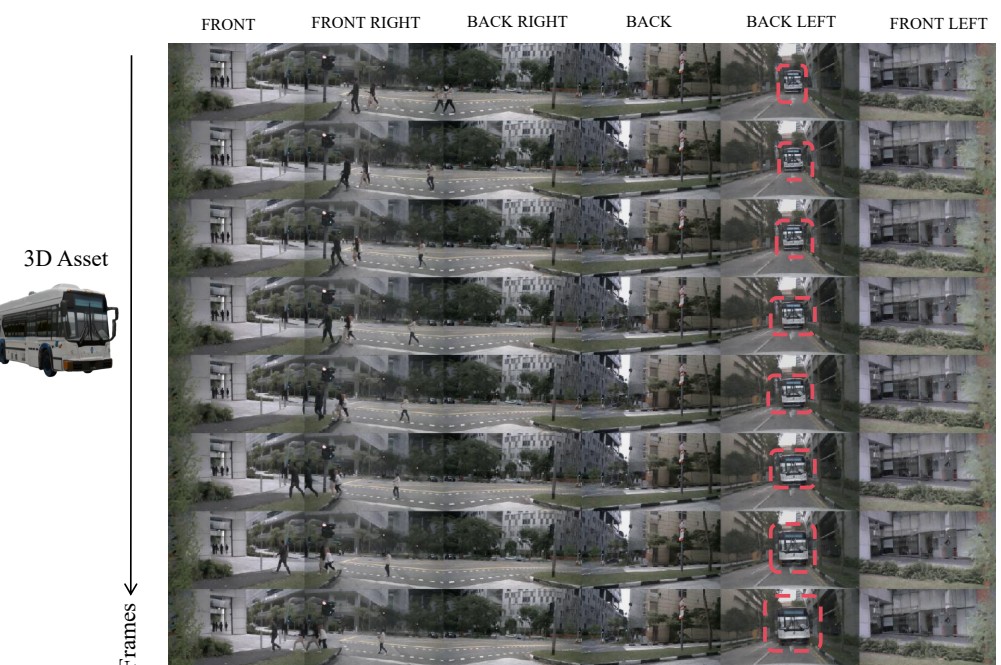

Figure 14: Insertion of a bus in the back-side region.

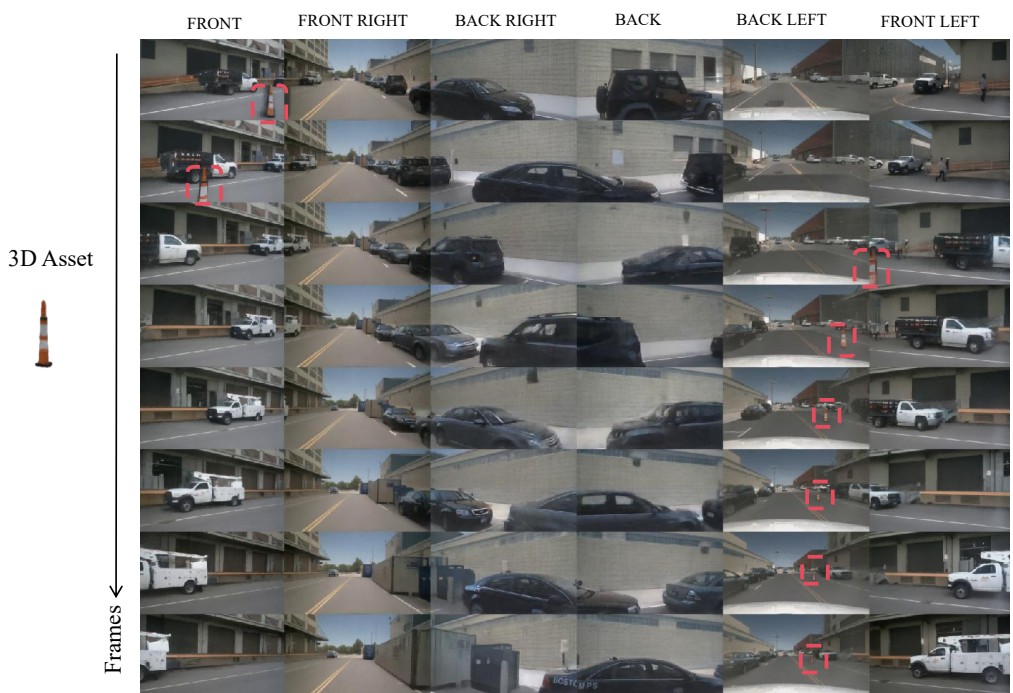

Figure 15: Insertion of a traffic cone in the left-side region.

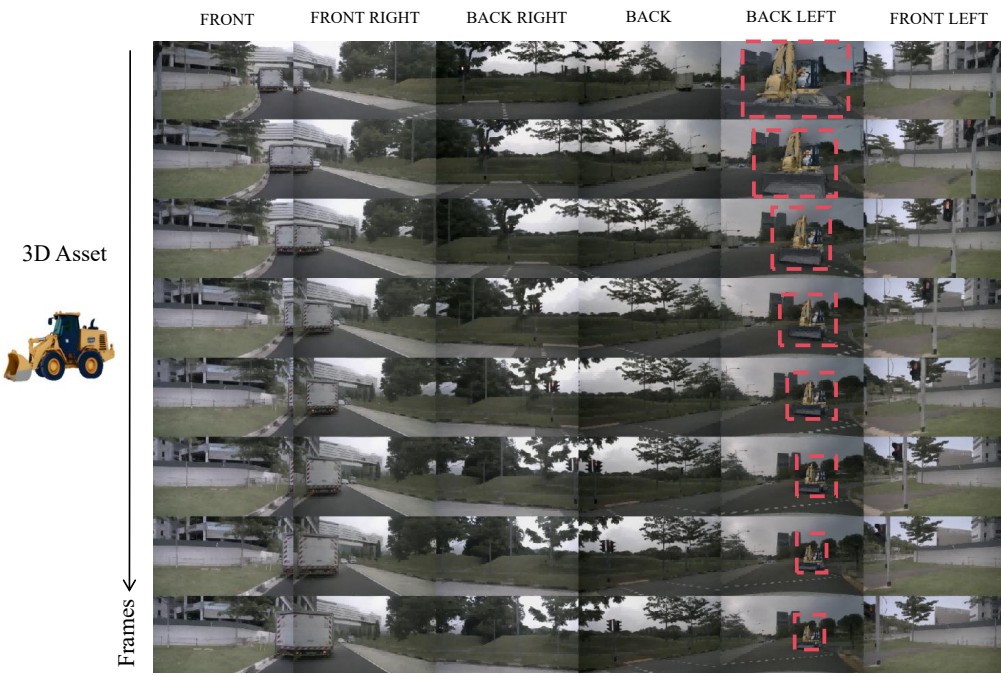

Figure 16: Insertion of a construction vehicle in the back-side region.

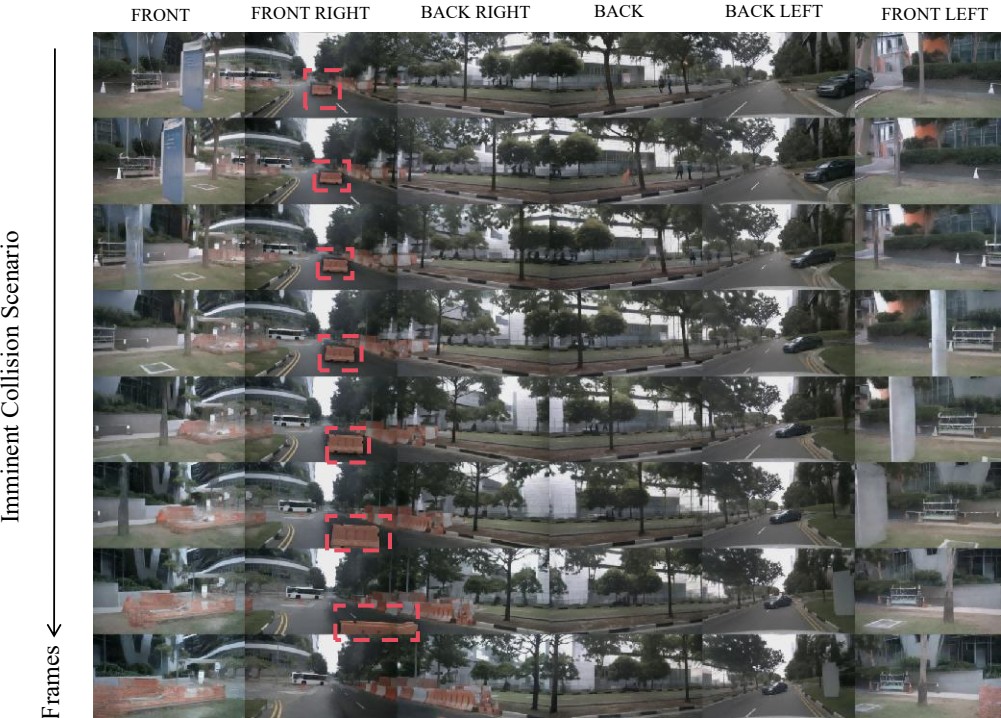

Figure 17: Corner case. Insertion of a barrier in the front-side region, where the ego vehicle is about to collide with it.

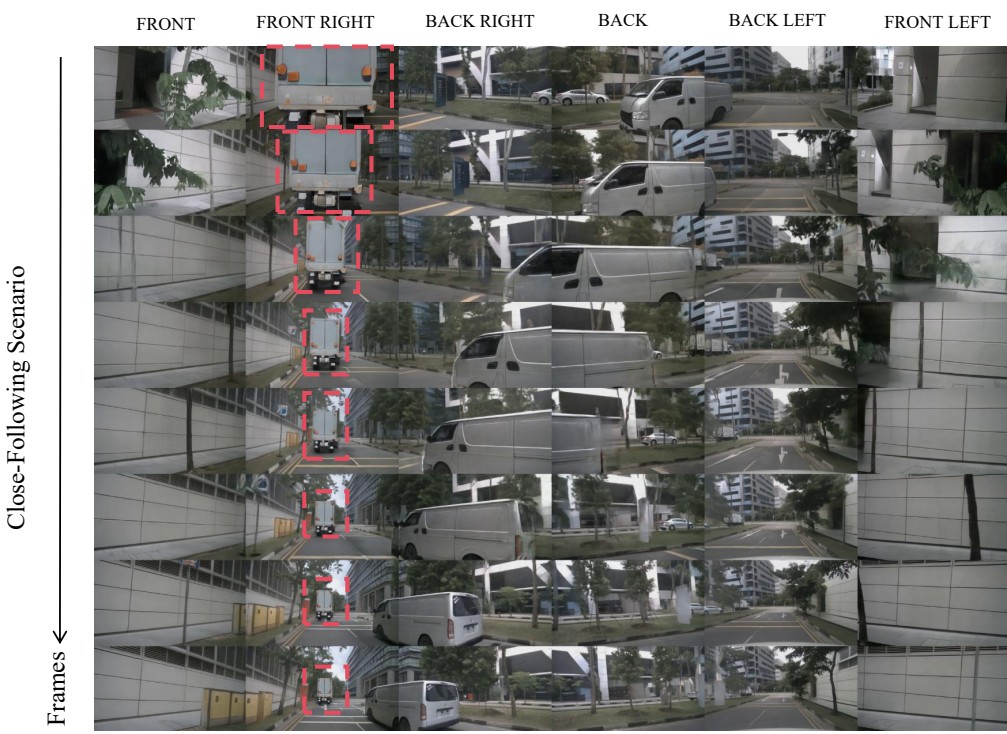

Figure 18: Corner case. Insertion of a truck in the close-range front-side region.

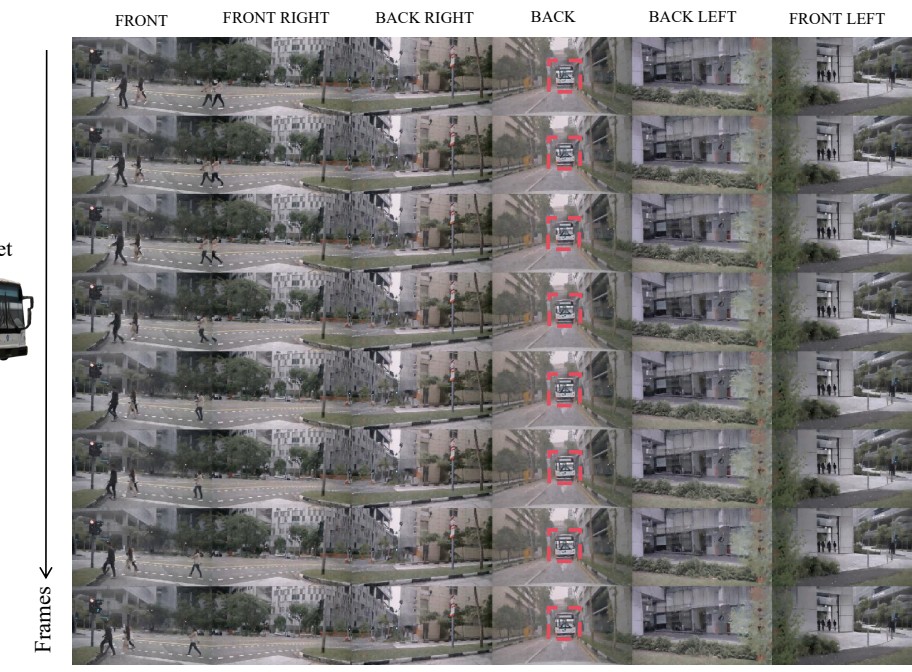

Figure 19: Style transfer. Insertion of a bus in the rain.

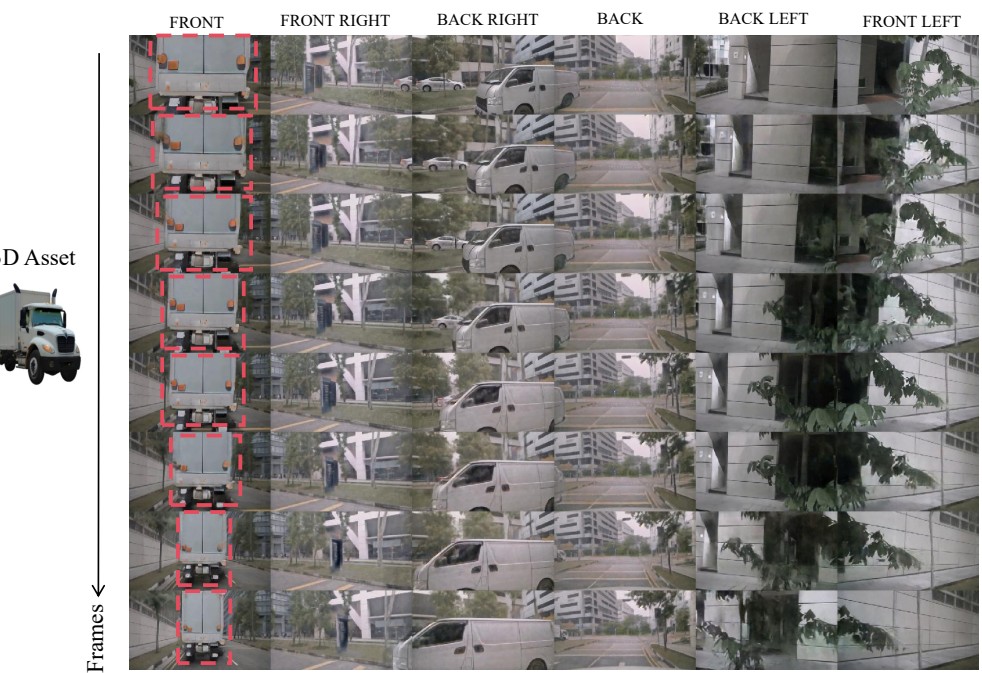

Figure 20: Style transfer. Insertion of a truck in the rain.

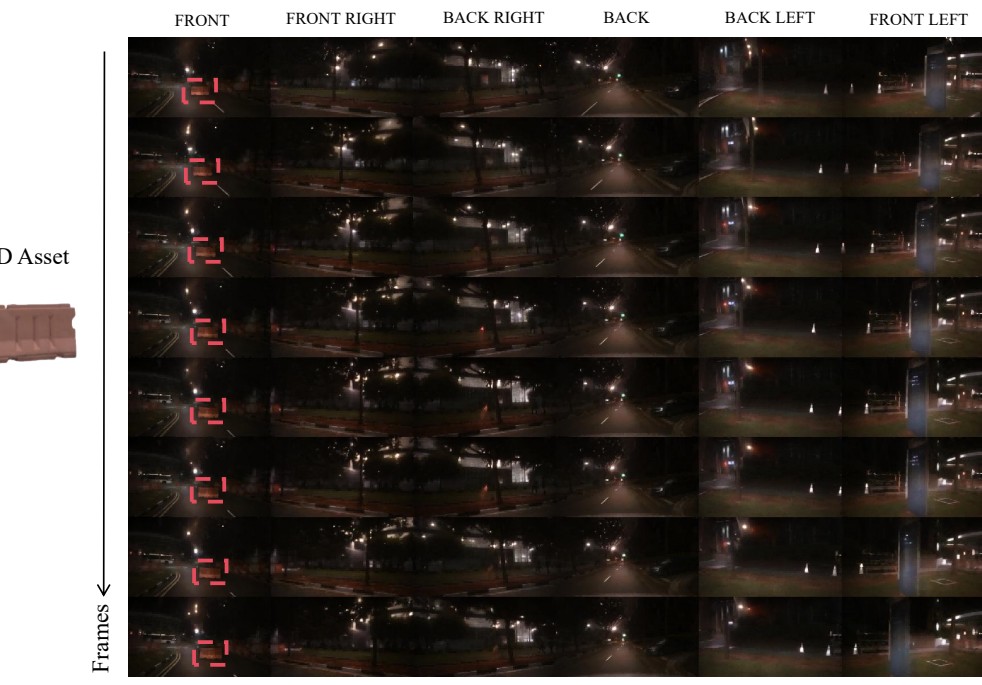

Figure 21: Style transfer. Insertion of a barrier in the night.

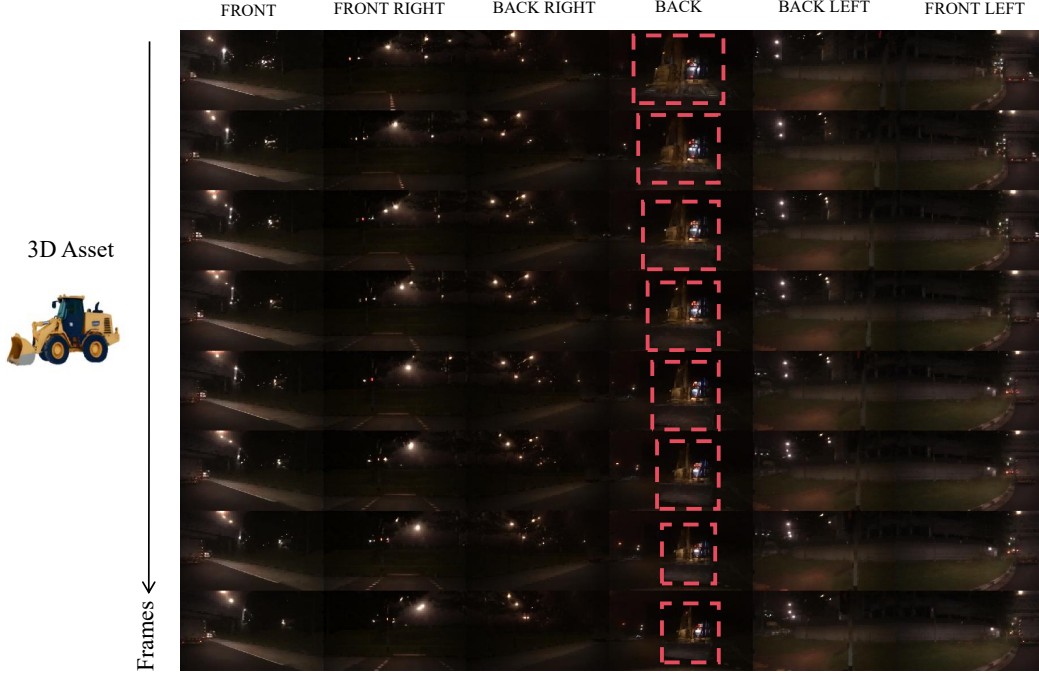

Figure 22: Style transfer. Insertion of a constructive vehicle in the night.

