# OpenReview forum: "Rethinking Driving World Model as Synthetic Data Generator for Perception Tasks"
_ICLR.cc/2026/Conference — ICLR 2026 Poster_

### Official Review · Reviewer_yxwm · 2025-10-16

**Soundness:** 3
**Presentation:** 2
**Contribution:** 2
**Rating:** 4
**Confidence:** 4

**Summary:**

This paper critically re-evaluates the utility of synthetic data generated by driving world models for downstream perception tasks. The authors argue that previous works often rely on an unfair evaluation protocol where models trained on hybrid (real + synthetic) data use twice the training epochs of real-data-only baselines. They demonstrate that when training epochs are matched, the benefits of existing synthetic data augmentation methods become negligible or even negative.

To address this, the paper introduces **Dream4Drive**, a novel framework for generating high-quality, 3D-aware synthetic data. Instead of relying on sparse conditioning signals like BEV maps, Dream4Drive first decomposes a real video into a set of dense, 3D-aware guidance maps (depth, normal, edge, etc.). It then renders 3D assets into these maps and fine-tunes a Diffusion Transformer-based world model to generate photorealistic, multi-view videos that incorporate these new objects. This approach allows for precise, instance-level control and ensures geometric and visual consistency.

Furthermore, the authors contribute **DriveObj3D**, a large-scale 3D asset dataset tailored for driving scenarios, along with an automated pipeline for its creation. Through extensive experiments on detection and tracking tasks, the paper shows that augmenting the training set with a very small fraction (<2%) of data generated by Dream4Drive consistently and significantly improves perception performance, even under fair, epoch-matched comparisons.

**Strengths:**

1. **Important and Timely Critique:** The central argument regarding the unfair evaluation of synthetic data is a significant contribution. By demonstrating that simply increasing training epochs on real data can match or exceed the performance of hybrid-data training, the paper forces the community to reconsider how the value of synthetic data is measured. This provides a strong and compelling motivation for the proposed work.
2. **Comprehensive and Convincing Experiments:** The experimental validation is thorough and directly supports the paper's claims.
    - The head-to-head comparisons under 1x, 2x, and 3x epoch settings provide clear evidence for the effectiveness of Dream4Drive over baselines.
    - The ablation studies are insightful, systematically analyzing the impact of insertion position, distance, 3D asset source, and different components of the guidance maps. This provides a deeper understanding of what makes the synthetic data effective.
    - The evaluation on both detection and tracking tasks, at multiple resolutions, demonstrates the general applicability of the generated data.

**Weaknesses:**

1. **Limited Scope of Generated Corner Cases:** The paper defines "corner cases" primarily as the insertion of new objects into a scene. While this is an important class of long-tail events, it does not cover other critical scenarios, such as adverse weather conditions (heavy rain, snow, fog), unusual lighting (lens flare, low sun), complex multi-agent interactions, or environmental changes (e.g., road construction). The framework's applicability to these other types of corner cases is not explored.
2. **Potential Bottlenecks in the Asset Generation Pipeline:** The `DriveObj3D` pipeline is a cascade of multiple sophisticated models (segmentation, multi-view generation, mesh generation). The final asset quality is contingent on the successful execution of every step. This pipeline may be brittle; for instance, a failure in multi-view generation could lead to an incomplete or distorted 3D mesh. A discussion on the robustness, failure modes, and potential need for manual curation of this pipeline would strengthen the paper.
3. **Limited Technical Contribution.** While the specific application and the system-level design are effective in research application, the work does not introduce a new core generative modeling technique or a new research framework. For some reviewers, the novelty might be perceived as incremental, lying more in the clever combination of existing parts than in foundational innovation.
4. **Computational Cost Analysis is Missing:** The proposed data generation process is multi-staged and involves fine-tuning a large Diffusion Transformer model. This suggests a significant computational cost. While the paper argues for fairness in terms of training *epochs*, a discussion on fairness in terms of total *compute* (cost of data synthesis + cost of training) would provide a more complete picture. For instance, how does the cost of generating 420 synthetic samples compare to the cost of training the perception model for an additional epoch?

**Questions:**

1. Regarding the `DriveObj3D` creation pipeline: Could you quantify the robustness of this pipeline? What is the approximate failure rate, and what are the common failure modes? How much manual filtering or intervention was required to curate the final high-quality dataset?
2. The scene selection for insertion is described as choosing frames "where no other vehicles appear along the insertion trajectory." How is this selection performed (manually or automatically)? Could this process introduce a bias, for example, by preferentially selecting less cluttered scenes, thereby limiting the complexity of the generated data?
3. The paper highlights improved realism via shadows and reflections. Does the generative model also learn to synthesize more complex physical interactions? For example, does an inserted vehicle generate splashes when moving through a puddle, or create dust on a dirt road? To what extent can the model handle nuanced lighting effects beyond direct shadows, such as colored light from traffic signals reflecting on the vehicle?

---

> ### Author Response · Authors · 2025-11-20
> **Response to Reviewer yxwm 1/2**
>
> We thank the reviewer for their thoughtful feedback! We address specific concerns and questions below.
>
> ------
>
> > **W1.** Limited Scope of Generated Corner Cases.
>
> - First, the core insight of our work is to **highlight unfair evaluation protocols for** **synthetic data** and demonstrate that **edited synthetic data improves downstream perception models**. Using **<2% additional data**, we achieve substantial gains (mAP: 36.1 → 40.7, NDS: 47.9 → 52.0). The focus is **not** on improving the generative model’s generalization to create extreme scenarios.
> - Second, Dream4Drive is a flexible video editing framework capable of modifying any autonomous-driving scene and inserting desired assets.
> - Additionally, style transfer can convert commonly edited scenes into rare weather or lighting conditions. Preliminary experiments show that incorporating style-transferred data further improves downstream performance (see Reviewer yb3h’s Q3).
>
> |        | 1x Epochs      |                             | 2x Epochs      |                             | 3x Epochs      |                             |
> | ------ | -------------- | --------------------------- | -------------- | --------------------------- | -------------- | --------------------------- |
> |        | Original (420) | Style Transfer（420 + 420） | Original (420) | Style transfer（420 + 420） | Original (420) | Style transfer（420 + 420） |
> | mAP ↑  | 40.7           | 41.2                        | 43.6           | 44.2                        | 44.5           | 44.8                        |
> | mATE ↓ | 64.2           | 63.7                        | 61.6           | 61.2                        | 59.8           | 59.7                        |
> | mAOE ↓ | 48.0           | 47.5                        | 39.4           | 39.0                        | 40.1           | 39.8                        |
> | mAVE ↓ | 27.1           | 26.8                        | 27.4           | 26.8                        | 27.2           | 26.9                        |
> | NDS ↑  | 52.0           | 52.4                        | 54.3           | 54.7                        | 55.0           | 55.2                        |
>
> - Finally, for complex multi-agent interactions, a multimodal LLM can annotate multiple safe trajectories, enabling Dream4Drive to insert multiple assets simultaneously and generate more complex multi-agent scenarios.
>
> ------
>
> > **W2. & Q1.** A discussion on the robustness, failure modes, and potential need for manual curation of this pipeline would strengthen the paper.
>
> - We evaluate synthesized asset quality via image similarity metrics (Figure 6),  with average asset similarity above 90% in CLIP IS, demonstrating pipeline robustness.
>
> |         |           | car    | truck  | bus    | trailer | construction vehicle | pedestrian | motorcycle | bicycle | traffic cone | barrier |
> | ------- | --------- | ------ | ------ | ------ | ------- | -------------------- | ---------- | ---------- | ------- | ------------ | ------- |
> |         | Trellis   | 0.8913 | 0.9160 | 0.8930 | 0.9041  | 0.8870               | 0.8753     | 0.9045     | 0.9273  | 0.9254       | 0.9270  |
> | CLIP IS | Hunyuan3D | 0.8929 | 0.9043 | 0.9031 | 0.9022  | 0.8787               | 0.8863     | 0.9121     | 0.9170  | 0.9288       | 0.9101  |
> |         | Ours      | 0.9048 | 0.9291 | 0.9197 | 0.9270  | 0.8996               | 0.9061     | 0.9201     | 0.9351  | 0.9305       | 0.9310  |
> |         |           |        |        |        |         |                      |            |            |         |              |         |
> |         | Trellis   | 0.7008 | 0.7266 | 0.7012 | 0.6974  | 0.6852               | 0.6721     | 0.6990     | 0.7053  | 0.7275       | 0.7304  |
> | DINO IS | Hunyuan3D | 0.7360 | 0.7631 | 0.7365 | 0.7344  | 0.7204               | 0.7074     | 0.7369     | 0.7405  | 0.7610       | 0.7652  |
> |         | Ours      | 0.8117 | 0.7979 | 0.8082 | 0.8040  | 0.7960               | 0.7835     | 0.8065     | 0.8129  | 0.8365       | 0.8394  |
>
> - The most common failure occurs in the first step when the asset screenshot is incomplete, which can mislead multi-view image generation and 3D mesh creation, producing incorrect sizes or failed assets.
> - The asset generation process **requires no manual effort**. Incomplete or failed assets are automatically filtered via similarity computation, and the remaining assets are normalized in width, height, and depth, producing a high-quality 3D asset dataset.
>
> ------
>
> > **W3.** Limited Technical Contribution.
>
> For a detailed discussion of the novelty and technical contributions, please refer to **General Response 2**.

---

> > ### Author Response · Authors · 2025-11-20
> > **Response to Reviewer yxwm 2/2**
> >
> > > **W4.** Computational Cost Analysis is Missing.
> >
> > - Negligible compute cost of the synthetic data pipeline: Our model is based on MagicDriveDiT and fine-tuned for 8 hours on 8×H200 GPUs (2,000 iterations). Once fine-tuned, generating edited video data is lightweight: 7 scenes × 10 assets × 10 steps × 4 s ≈ **50 minutes** of GPU time. In contrast, training the downstream perception model on 28,130 real samples takes ≈**13 hours**, dominating the computational budget. Adding 420 synthetic samples adds negligible overhead. Overall, the synthetic pipeline’s compute cost is a small fraction of end-to-end training.
> > - Synthetic data provides gains beyond additional epochs: Increasing training epochs mainly reduces optimization error, whereas synthetic data improves **distribution coverage**, especially for rare cases. Using only real data, the model achieves 36.1 → 42.2 → 43.1 mAP over 1–3 epochs. Adding **<2% synthetic samples** increases this to 40.7 → 43.6 → 44.5 mAP (+4.6, +1.4, +1.4), demonstrating that synthetic data **enhances performance beyond mere epoch increases and extends model generalization**.
> >
> > ------
> >
> > > **Q2.** A discussion on the scene selection and potential bias.
> >
> > - We use an MLLM-assisted pipeline to select scenes. The model identifies whether a continuous safe driving trajectory exists in any direction (front, back, left, right) and outputs the initial asset insertion position (x, y, z in the ego-vehicle frame at the first frame). Detailed prompts are provided in the Appendix A.2.
> > - Following the reviewer’s suggestion, we randomly selected two alternative sets of insertion scenes, forming three groups (A, B, C) including the original. Downstream task experiments show similar performance across all groups, indicating that any bias from scene selection is negligible.
> >
> > |       | 1x Epochs       |                 |                 | 2x Epochs       |                 |                 | 3x Epochs       |                 |                 |
> > | ----- | --------------- | --------------- | --------------- | --------------- | --------------- | --------------- | --------------- | --------------- | --------------- |
> > |       | Insert Scenes A | Insert Scenes B | Insert Scenes C | Insert Scenes A | Insert Scenes B | Insert Scenes C | Insert Scenes A | Insert Scenes B | Insert Scenes C |
> > | mAP↑  | 40.7            | 40.8            | 40.7            | 43.6            | 43.5            | 43.6            | 44.5            | 44.3            | 44.5            |
> > | mATE↓ | 64.2            | 64.1            | 64.2            | 61.6            | 61.5            | 61.4            | 59.8            | 59.8            | 59.9            |
> > | mAOE↓ | 48.0            | 48.0            | 48.1            | 39.4            | 39.4            | 39.5            | 40.1            | 40.2            | 40.1            |
> > | mAVE↓ | 27.1            | 26.9            | 27.1            | 27.4            | 27.5            | 27.3            | 27.2            | 27.3            | 27.2            |
> > | NDS↑  | 52.0            | 52.1            | 52.1            | 54.3            | 54.2            | 54.3            | 55.0            | 55.0            | 55.1            |
> >
> > ------
> >
> > > **Q3.** Does the generative model also learn to synthesize more complex physical interactions?
> >
> > We thank the reviewer for this insightful question, which touches upon the current frontiers of generative modeling.
> >
> > Our approach prioritizes **semantic-level realism** by ensuring plausible appearance, correct shadows, and basic reflections, which is sufficient to enable robust feature learning and improve downstream performance.
> >
> > However, synthesizing complex physical interactions, such as dynamic splashes, dust clouds, or nuanced colored light reflections from traffic signals, falls **outside the scope** of our primary contribution. These effects require advanced physics-based rendering or highly sophisticated generative models (even the current SOTA diffusion-based models struggle with such fine-grained interactions).
> >
> > Crucially, **our method’s core contribution is identifying and synthesizing the valuable out-of-distribution scenarios for perception training**, not optimizing the underlying generative model to achieve perfect photorealism.
> >
> > -----
> >
> > Finally, thank you for your thoughts and valuable feedback! We hope we have addressed most of your concerns. If you have any questions, please don't hesitate to reach out to us.

---

> ### Author Response · Authors · 2025-11-27
> **Sorry to bother you**
>
> Dear reviewer yxwm,
>
> Thanks for the time and effort you've put into reviewing our submission. We have provided more explanations and answers to your questions.
>
> As the discussion deadline is approaching, we kindly invite you to share your thoughts and engage in discussion if possible. If there are any points you'd like us to clarify or discuss further, we are more than happy to assist.
>
> Thank you again for your valuable feedback and contributions.
>
> Best regards,
>
> Authors of Paper 971

---

### Official Review · Reviewer_yb3h · 2025-10-29

**Soundness:** 3
**Presentation:** 3
**Contribution:** 2
**Rating:** 4
**Confidence:** 4

**Summary:**

This paper proposes Dream4Drive, a framework for generating synthetic training data for autonomous driving perception tasks. The key insight is that previous work evaluated synthetic data unfairly by pretraining on synthetic data and then finetuning on real data (2x training epochs), obscuring true benefits. Dream4Drive decomposes videos into 3D-aware guidance maps (depth, normal, edge, object image, mask) and uses a fine-tuned diffusion transformer to render edited multi-view videos after inserting 3D assets. The authors contribute DriveObj3D, a dataset of 3D assets across driving scenarios. With only 420 synthetic samples (<2% of training data), they demonstrate improvements in detection and tracking at 1x, 2x, and 3x training epochs.

**Strengths:**

- The paper identifies a key shortcoming in most generative simulators for autonomous driving data: improvements in downstream task from training on both real and synthetic data have in the past been attributed only to the addition of synthetic data while ignoring that increasing training epochs improves performance even just using the base dataset (nuScenes).
- The paper provides a large-scale dataset (DriveObj3D) along with a method to insert assets from the dataset in a natural manner to match scene lighting and shadows.
- The results show some improvement over existing methods in downstream detection tasks while being sample efficient (only adding <2% data over the original dataset)

**Weaknesses:**

W1. The paper lacks novelty beyond the Epoch related finding. The actual data generation framework is a composition of existing models and methods rather than a technical contribution. While adding multiview images as conditions to generate a 3D mesh for 3D assets does improve asset quality over meshes generated from single view, it seems like an unsurprising improvement. Moreover, the actual improvement in scores from Table 5 compared to other 3D asset generation methods is relatively small, despite using multiview images vs Hunyuan3D's single view method.

W2. The detection performance results using the method are marginal improvements over previous works. In Tables 1, 2, and 3, the relative improvement in metrics like mAP in the 2x and 3x Epochs are small, even over just using the original nuScenes dataset. In fact, this result shows that even the DriveObj3D falls prey to the core discovery the authors note about normalizing for # of Epochs trained; the improvement over training with only real data diminishes with increasing Epochs. The increase in performance in mAP vs Real data in Table 1 (2x Epochs) is <1% improvement despite including >1% increase in training data.

W3. Appendix E acknowledges that "automatically ensuring insertions are in drivable areas and avoid collisions remains an open challenge." This is a significant limitation, with manual effort required preventing this method from generating corner cases on a significant scale.

**Questions:**

Additional questions:

Q1: Given that naive insertion performs within 0.6-0.7 mAP of your full method (Tables 3-4), can you provide a compute or latency cost-benefit analysis? The qualitative improvement in image quality is clear (Figure 8) but the actual metric benefits less so.

Q2: How much manual effort is required to select 420 valid insertion positions and trajectories? Can you quantify the human time required and discuss path to full automation?

Q3: Building off Q2, do the authors have an evidence or intuition as to whether the approach will scale with more augmented scenes using their library? W2 suggests that the increase in training data doesn't necessarily correspond with a proportional increase in performance.

---

> ### Author Response · Authors · 2025-11-20
> **Response to Reviewer yb3h 1/3**
>
> Thank you for the detailed and constructive feedback! We treasure the opportunity to address your concerns and improve our work.
>
> ------
>
> > **W1.** Novelty and technical contribution. The actual improvement in scores from Table 5 compared to other 3D asset generation methods is relatively small.
>
> - For a detailed discussion of the novelty and technical contribution, please refer to **General Response 2**.
>
> - Clarifying the Contribution of DriveObj3D
>
>   - Classic 3D detection methods [1,2,3] yield **only 1–2%** **NDS** **gain** over prior SOTA. Previous data augmentation approaches [4, 5], which double the training data, actually **reduce NDS by 1.2%** (50.4 → 49.2, Table 1).
>
>   - Our 3D asset synthesis method, using **<2% additional data**, **improves NDS by 1.2%** over ordinary asset augmentation (50.8 → 52.0, Table 5).
>
>   - We further evaluate synthesized asset quality via image similarity metrics (Figure 6), showing that our pipeline consistently outperforms all baselines across all metrics.
>
>   - |         |           | car    | truck  | bus    | trailer | construction vehicle | pedestrian | motorcycle | bicycle | traffic cone | barrier |
>       | ------- | --------- | ------ | ------ | ------ | ------- | -------------------- | ---------- | ---------- | ------- | ------------ | ------- |
>       |         | Trellis   | 0.8913 | 0.9160 | 0.8930 | 0.9041  | 0.8870               | 0.8753     | 0.9045     | 0.9273  | 0.9254       | 0.9270  |
>       | CLIP IS | Hunyuan3D | 0.8929 | 0.9043 | 0.9031 | 0.9022  | 0.8787               | 0.8863     | 0.9121     | 0.9170  | 0.9288       | 0.9101  |
>       |         | Ours      | 0.9048 | 0.9291 | 0.9197 | 0.9270  | 0.8996               | 0.9061     | 0.9201     | 0.9351  | 0.9305       | 0.9310  |
>       |         |           |        |        |        |         |                      |            |            |         |              |         |
>       |         | Trellis   | 0.7008 | 0.7266 | 0.7012 | 0.6974  | 0.6852               | 0.6721     | 0.6990     | 0.7053  | 0.7275       | 0.7304  |
>       | DINO IS | Hunyuan3D | 0.7360 | 0.7631 | 0.7365 | 0.7344  | 0.7204               | 0.7074     | 0.7369     | 0.7405  | 0.7610       | 0.7652  |
>       |         | Ours      | 0.8117 | 0.7979 | 0.8082 | 0.8040  | 0.7960               | 0.7835     | 0.8065     | 0.8129  | 0.8365       | 0.8394  |
>
> ------

---

> > ### Author Response · Authors · 2025-11-20
> > **Response to Reviewer yb3h 2/3**
> >
> > > **W3. & Q2.** The method lacks automation. How much manual effort is required to select 420 valid insertion positions and trajectories and can you discuss path to full automation?
> >
> > - We apologize for any confusion. The **General Response 1** already details the full data generation process and time costs. While not fully automated, our method requires **minimal manual effort**—for instance, synthesizing all 420 samples takes **only 295 seconds of human intervention**, making it easily scalable.
> > - Potential Automation Pathways.
> >   - MLLM-based annotation. Fine-tune a multimodal large language model to determine whether a drivable region exists, and output the initial insertion position and corresponding orientation of the asset.
> >   - Occupancy-grid modeling. By generating an occupancy grid from the original video, drivable regions can be clearly identified. Given a start and end position, a path-planning algorithm such as A* can then find a safe trajectory.
> >   - Traditional lane-detection approaches. By detecting lane and center lines, one can infer the directions of drivable lanes, and automatically determine feasible asset insertion regions and their yaw orientation.
> >
> > ------
> >
> > > **Q3.** Do the authors have an evidence or intuition as to whether the approach will scale with more augmented scenes using their library?
> >
> > We sincerely thank the reviewer for the constructive suggestion.
> >
> > - As addressed in W3 and Q2, our method is easily scalable.
> > - Following your suggestion, we conduct scaling experiments to assess the effect of increasing synthetic data. Results (table below) show no significant scaling benefit. In fact, excessive OOD data slightly decreases downstream performance. We hypothesize that small amounts of OOD data improve robustness (Table 3), whereas too much dilutes in-distribution features.
> >
> > |        | 1x Epochs |           |           | 2x Epochs |           |           | 3x Epochs |           |           |
> > | ------ | --------- | --------- | --------- | --------- | --------- | --------- | --------- | --------- | --------- |
> > |        | 7 Scenes  | 14 Scenes | 35 Scenes | 7 Scenes  | 14 Scenes | 35 Scenes | 7 Scenes  | 14 Scenes | 35 Scenes |
> > | mAP ↑  | 40.7      | 40.4      | 39.7      | 43.6      | 43.1      | 42.3      | 44.5      | 44.1      | 43.6      |
> > | mATE ↓ | 64.2      | 64.4      | 64.5      | 61.6      | 62.0      | 62.5      | 59.8      | 59.5      | 60.3      |
> > | mAOE ↓ | 48.0      | 49.7      | 51.6      | 39.4      | 39.8      | 40.2      | 40.1      | 40.3      | 40.4      |
> > | mAVE ↓ | 27.1      | 27.8      | 28.4      | 27.4      | 28.1      | 28.4      | 27.2      | 27.7      | 27.6      |
> > | NDS ↑  | 52.0      | 51.6      | 50.9      | 54.3      | 53.8      | 53.1      | 55.0      | 54.6      | 54.2      |
> >
> > - To examine whether more OOD data yields greater gains,  we apply style transfer to new synthetic samples (e.g., rain or nighttime conditions) and combine them with the original data for training. This further improves downstream performance.
> >
> > |        |   1x Epochs    |                             | 2x Epochs      |                             | 3x Epochs      |                             |
> > | ------ | :------------: | --------------------------- | -------------- | --------------------------- | -------------- | --------------------------- |
> > |        | Original (420) | Style Transfer（420 + 420） | Original (420) | Style transfer（420 + 420） | Original (420) | Style transfer（420 + 420） |
> > | mAP ↑  |      40.7      | 41.2                        | 43.6           | 44.2                        | 44.5           | 44.8                        |
> > | mATE ↓ |      64.2      | 63.7                        | 61.6           | 61.2                        | 59.8           | 59.7                        |
> > | mAOE ↓ |      48.0      | 47.5                        | 39.4           | 39.0                        | 40.1           | 39.8                        |
> > | mAVE ↓ |      27.1      | 26.8                        | 27.4           | 26.8                        | 27.2           | 26.9                        |
> > | NDS ↑  |      52.0      | 52.4                        | 54.3           | 54.7                        | 55.0           | 55.2                        |
> >
> > - **Conclusion:** We find that downstream models benefit from OOD **scene layouts and diverse environmental conditions**, rather than merely replicating real data. Visualization examples of new synthetic data are included in the revised version.

---

> > > ### Author Response · Authors · 2025-11-20
> > > **Response to Reviewer yb3h 3/3**
> > >
> > > > **W2.** Small gains from Dream4Drive with increasing epochs, and why 1% more data does not yield 1% improvement.
> > >
> > > - First, we clarify the relative improvements in metrics like mAP across epochs. Prior data augmentation methods adding a similar amount of data under fair comparison actually **decreased mAP**. In contrast, our method achieves meaningful gains with **<2% extra data**: first epoch +4.6 (36.1 → 40.7), second +1.4 (42.2 → 43.6), third +1.4 (43.1 → 44.5).
> > > - Second, it is natural for performance gains to slow down in later training epochs, as most models gradually converge toward the end of training. The baseline model's performance gain from **2x to 3x epochs is only +0.9** (42.2→ 43.1), which already indicates that the model is approaching its **convergence limit** on the original nuScenes data. Our method, even at this highly converged stage, still provides a consistent **+1.4 absolute mAP improvement** in both the 2x and 3x settings.
> > > - Finally, we respectfully clarify the interpretation regarding the data-to-performance ratio. It is a known characteristic of complex deep learning tasks, such as 3D object detection on nuScenes, that performance gains follow the **law of diminishing returns**, meaning a  ≥1% data increase is not expected to yield a ≥1% mAP gain, especially when the baseline is already highly optimized. The value of our contribution lies not just in the quantity (≤ 2% additional data), but in the **quality and novelty**. This synthetic data specifically targets hard, challenging scenarios, acting as a powerful regularizer to push the model beyond its convergence limits on real data. Therefore, achieving a consistent **absolute gain of  ～+1.4 mAP** is an **efficient and meaningful improvement** when considering the minimal cost and high quality of the injected data.
> > >
> > > ------
> > >
> > > > **Q1.** Given that naive insertion performs within 0.6-0.7 mAP of your full method (Tables 3-4), can you provide a compute or latency cost-benefit analysis? The qualitative improvement in image quality is clear (Figure 8) but the actual metric benefits less so.
> > >
> > > We thank the reviewer for the suggestion.
> > >
> > > - First, our method differs from naive insertion only by the **DiT re-rendering step**. Synthesizing 420 training samples involves 70 videos, 10 inference steps per video, each taking 4 seconds on an H200 GPU—totaling approximately **50 minutes**. Detailed timing is provided in the **General Response 1**.
> > > - Second, regarding the comment that the actual metric benefits are limited: as shown in Table 1, method [4] **doubled the training data**, and in our reproduction the data synthesis alone took about 12 hours, yet it **reduced mAP by 1.3** (38.4 → 37.1). In contrast, our approach achieves a **0.6–0.7 mAP improvement** with only **50 minutes** of processing. This demonstrates that our method provides superior **efficiency and effectiveness** compared with simply scaling up the dataset size.
> > >
> > > ------
> > >
> > > [1] BEVFormer: Learning Bird’s-Eye-View Representation from Multi-Camera Images via Spatiotemporal Transformers. ECCV 2022.
> > >
> > > [2] Exploring Object-Centric Temporal Modeling for Efficient Multi-View 3D Object Detection. ICCV 2023.
> > >
> > > [3] BEVFusion: Multi-Task Multi-Sensor Fusion with Unified Bird’s-Eye View Representation. ICRA 2023.
> > >
> > > [4] Panacea: Panoramic and Controllable Video Generation for Autonomous Driving. CVPR 2024.
> > >
> > > [5] SubjectDrive: Scaling Generative Data in Autonomous Driving via Subject Control. AAAI 2025.
> > >
> > > -----
> > >
> > > Finally, thank you for your thoughts and valuable feedback! We hope we have addressed most of your concerns. If you have any questions, please don't hesitate to reach out to us.

---

### Official Review · Reviewer_FTf2 · 2025-10-31

**Soundness:** 2
**Presentation:** 2
**Contribution:** 3
**Rating:** 6
**Confidence:** 4

**Summary:**

The paper presents Dream4Drive, a 3D-aware synthetic data generation framework designed to improve downstream perception tasks such as 3D object detection and tracking in autonomous driving. Dream4Drive employs dense 3D-aware guidance maps to edit real driving videos by rendering high-quality 3D assets, ensuring multi-view geometric consistency and photorealism. A new dataset, DriveObj3D, is introduced to provide diverse 3D vehicle and traffic assets for large-scale, geometry-consistent scene editing. Dream4Drive demonstrates that injecting fewer than 2% synthetic samples consistently improves detection and tracking metrics, outperforming existing augmentation baselines.

**Strengths:**

1. Dream4Drive introduces a combination of dense 3D-aware guidance maps with generative video editing, enabling realistic, geometry-consistent, multi-view scene synthesis.

2. Demonstrating that fewer than 2% synthetic samples can enhance real data training is a strong and practically relevant result, suggesting high efficiency in data augmentation.

3. The proposed DriveObj3D fills a gap in open 3D asset resources for driving research, supporting reproducibility, and future extension in video-level generative modeling.

**Weaknesses:**

1. The paper does not clarify how inserted 3D assets are constrained to remain physically valid: ensuring no collisions, adherence to drivable areas, and correct orientations, nor how occlusion layers between objects are resolved across depth, normal, and edge maps.

2. There is no qualitative criterion for evaluating 3D asset realism or consistency, raising uncertainty about the reliability of the generated dataset and edited scenes.

3. As shown in Table 3, the method’s performance fluctuates across epochs; in some settings, direct insertion outperforms Dream4Drive, and differences between methods are within statistical noise, suggesting the need for repeated trials with mean–variance reporting.

4. The introduction is verbose, and the method section lists procedural steps without sufficient motivation or justification. Figures contain minor layout inconsistencies.

5. Missing Related Work: SimGen (NeurIPS 2024).

**Questions:**

Please refer to the weaknesses above

---

> ### Author Response · Authors · 2025-11-20
> **Response to Reviewer FTf2 1/2**
>
> Thank you for the detailed and constructive feedback! We treasure the opportunity to address your concerns and improve our work.
>
> ------
>
> > **W1.** The paper does not clarify how inserted 3D assets are constrained to remain physically valid.
>
> Thank you very much for your reminder. Here is the detailed explanation of the asset insertion process.
>
> - Drivable and collision-free placement. An MLLM identifies scenes with at least one valid collision-free trajectory, and all selected scenes are manually verified for physically valid asset placement.
> - Orientation correctness. Using *pyrender* and nuScenes camera parameters, we calibrate yaw angles for each scene. Assets are pre-standardized, so orientation is annotated only once per scene, with minimal manual effort (see **General Response 1**).
> - Occlusion and guidance maps editing. The insertion path is object-free, preventing occlusion. Depth, normal, and edge maps are updated by removing the inserted-asset region, and corresponding edits are applied to object patches and masks. The fine-tuned DiT model then generates geometrically consistent insertions guided by the edited guidance maps.
>
> ------
>
> > **W2.** There is no qualitative criterion for evaluating 3D asset realism or consistency, raising uncertainty about the reliability of the generated dataset and edited scenes.
>
> Thanks for your suggestion. We evaluate synthesized 3D assets using image similarity metrics against the originals (Figure 6).
>
> |         |           | car    | truck  | bus    | trailer | construction vehicle | pedestrian | motorcycle | bicycle | traffic cone | barrier |
> | ------- | --------- | ------ | ------ | ------ | ------- | -------------------- | ---------- | ---------- | ------- | ------------ | ------- |
> |         | Trellis   | 0.8913 | 0.9160 | 0.8930 | 0.9041  | 0.8870               | 0.8753     | 0.9045     | 0.9273  | 0.9254       | 0.9270  |
> | CLIP IS | Hunyuan3D | 0.8929 | 0.9043 | 0.9031 | 0.9022  | 0.8787               | 0.8863     | 0.9121     | 0.9170  | 0.9288       | 0.9101  |
> |         | Ours      | 0.9048 | 0.9291 | 0.9197 | 0.9270  | 0.8996               | 0.9061     | 0.9201     | 0.9351  | 0.9305       | 0.9310  |
> |         |           |        |        |        |         |                      |            |            |         |              |         |
> |         | Trellis   | 0.7008 | 0.7266 | 0.7012 | 0.6974  | 0.6852               | 0.6721     | 0.6990     | 0.7053  | 0.7275       | 0.7304  |
> | DINO IS | Hunyuan3D | 0.7360 | 0.7631 | 0.7365 | 0.7344  | 0.7204               | 0.7074     | 0.7369     | 0.7405  | 0.7610       | 0.7652  |
> |         | Ours      | 0.8117 | 0.7979 | 0.8082 | 0.8040  | 0.7960               | 0.7835     | 0.8065     | 0.8129  | 0.8365       | 0.8394  |
>
> Our pipeline consistently outperforms all baselines, and Table 5 shows that these assets lead to superior downstream perception improvements compared with alternative methods.

---

> > ### Author Response · Authors · 2025-11-20
> > **Response to Reviewer FTf2 2/2**
> >
> > > **W3.**  The method’s performance fluctuates across epochs, suggesting the need for repeated trials with mean–variance reporting.
> >
> > We sincerely thank the reviewer for the feedback and suggestions.
> >
> > - For **mAP, mATE, and NDS**, our method consistently improves with more training epochs and outperforms direct insertion. Metrics like **mAOE** and **mAVE** are less critical, as NDS better reflects overall downstream performance.
> > - Our method mainly enhances **visual realism** (Figure 8). Since insertion orientation and speed are identical between naive insert and ours, they should theoretically have similar effects on mAOE and mAVE; observed fluctuations likely stem from training noise.
> > - Following your suggestion, we repeated all experiments three additional times and report the mean and variance. The results show that mAOE and mAVE variations remain within the expected noise range.  Fluctuations are more pronounced in later training stages, likely because smaller parameter updates make the model more sensitive to minor perturbations, causing these sensitive metrics to vary, whereas early-stage updates largely absorb such effects.
> >
> > |        | 1x Epochs |              |          | 2x Epochs |              |          | 3x Epochs |              |          |
> > | ------ | --------- | ------------ | -------- | --------- | ------------ | -------- | --------- | ------------ | -------- |
> > |        | Real      | Naive Insert | Ours     | Real      | Naive Insert | Ours     | Real      | Naive Insert | Ours     |
> > | mAP ↑  | 36.1±0.0  | 40.1±0.0     | 40.7±0.0 | 42.2±0.1  | 42.9±0.2     | 43.6±0.2 | 43.1±0.1  | 43.1±0.2     | 44.5±0.2 |
> > | mATE ↓ | 69.2±0.0  | 64.7±0.1     | 64.2±0.0 | 61.6±0.1  | 62.2±0.2     | 61.8±0.2 | 60.5±0.1  | 61.2±0.3     | 60.1±0.3 |
> > | mAOE ↓ | 56.7±0.0  | 49.0±0.1     | 47.9±0.1 | 43.2±0.0  | 38.5±1.1     | 38.7±0.8 | 45.7±0.1  | 39.6±1.2     | 39.8±0.9 |
> > | mAVE ↓ | 28.5±0.0  | 28.4±0.1     | 27.1±0.1 | 27.5±0.0  | 27.3±0.3     | 27.3±0.3 | 27.4±0.0  | 27.3±0.3     | 27.3±0.3 |
> > | NDS ↑  | 47.9±0.0  | 51.3±0.1     | 52.0±0.0 | 53.2±0.0  | 53.8±0.3     | 54.5±0.2 | 53.6±0.1  | 54.2±0.3     | 54.9±0.3 |
> >
> > ------
> >
> > > **W4.** The introduction is verbose, and the method section lists procedural steps without sufficient motivation or justification. Figures contain minor layout inconsistencies.
> >
> > We appreciate the reviewer’s valuable suggestions. We have revised the Introduction and Method sections and ensured figure consistency. Please refer to the updated version for details:
> >
> > - **Introduction:** Condensed the related work to reduce redundancy and streamlined the method description.
> > - **Method:** Added the necessary motivation at the beginning of each section.
> > - **Figures:** Aligned the arrow directions in Figure 4 and corrected the blank regions previously present in the guidance map.
> >
> > ------
> >
> > > **W5. Missing Related Work.**
> >
> > We thank the reviewer for the reminder regarding relevant references, which have been incorporated. We clarify that both our work and SimGen emphasize the importance of **data appearance and layout diversity** for enhancing downstream perception models. However, our methodology differs: **SimGen** achieves diversity by expanding **control conditions** within simulation, whereas **our approach** achieves it by directly **editing existing real-world scenes** and inserting new assets to synthesize targeted out-of-distribution (OOD) scenarios.
> >
> > -----
> >
> > Finally, thank you for your thoughts and valuable feedback! We hope we have addressed most of your concerns. If you have any questions, please don't hesitate to reach out to us.

---

> ### Author Response · Authors · 2025-11-27
> **Sorry to bother you**
>
> Dear reviewer FTf2,
>
> Thanks for the time and effort you've put into reviewing our submission. We have provided more explanations and answers to your questions.
>
> As the discussion deadline is approaching, we kindly invite you to share your thoughts and engage in discussion if possible. If there are any points you'd like us to clarify or discuss further, we are more than happy to assist.
>
> Thank you again for your valuable feedback and contributions.
>
> Best regards,
>
> Authors of Paper 971

---

### Official Review · Reviewer_dArx · 2025-11-01

**Soundness:** 4
**Presentation:** 4
**Contribution:** 4
**Rating:** 8
**Confidence:** 4

**Summary:**

This paper primarily investigates the limitations of existing generation methods in downstream tasks, demonstrating that significant improvements can be achieved using only 2% of generated data. It also introduces DriveObj3D, a large-scale 3D asset dataset covering typical categories in driving scenarios and enabling diverse 3D-aware video editing.

**Strengths:**

- Identifies and analyzes the limitations of current generative methods in downstream tasks, showing that even a small amount (as little as 2%) of high-quality generated data can lead to significant performance gains.
- Introduces DriveObj3D, a large-scale 3D asset dataset encompassing typical object categories in driving scenarios, which supports diverse and 3D-aware video editing applications.

**Weaknesses:**

- The method relies on scene insertion and asset insertion that currently lack sufficient automation and are primarily dependent on manual effort.

**Questions:**

This work demonstrates the effectiveness of out-of-distribution (OOD) scene data generation for downstream task training, focusing specifically on *scene-level* OOD—namely, inserting vehicles into originally empty scenes. However, the scope of OOD scenarios could be further expanded. For instance, one could explore modifying existing scenes by altering vehicle trajectories or swapping vehicle categories to investigate how such diverse OOD variations impact downstream task performance.

---

> ### Author Response · Authors · 2025-11-20
> **Response to Reviewer dArx**
>
> We really appreciate the reviewer for the constructive comments and positive feedback on our paper.
>
> ------
>
> > **W1.** The method relies on scene insertion and asset insertion that currently lack sufficient automation and are primarily dependent on manual effort.
>
> We clarify that our pipeline is **semi-automated**, and the required human effort is **minimal**. Synthesizing all 420 samples requires only about **295 seconds of manual work**.
>
> |                             Step                             | Automation（s） | Manual effort（s） |
> | :----------------------------------------------------------: | --------------- | ------------------ |
> | Scene selection & initial position annotation (MLLM-assisted) | 700×10          |                    |
> |                      Scene verification                      |                 | 7×20               |
> |                 Scene orientation annotation                 |                 | 7×15               |
> |                       Asset selection                        |                 | 10×5               |
> |                    Video generation (DiT)                    | 70×10×4         |                    |
> |                            Total                             | 9800s ≈ 2.72h   | 295s               |
>
> A detailed analysis of the synthesis procedure and time cost is provided in the **General Response 1**.
>
> ------
>
> > **Q1.** Explore the impact of diverse OOD variations on downstream task performance.
>
> Following your suggestion, we conduct fine-grained ablations on **trajectory speed** and **asset categories**.
>
> - Trajectory speed: Inserted vehicles always move straight; we varied their speed across three levels (2 / 5 / 8 m/s). Higher speeds consistently yield larger downstream gains, aligning with Table 5 in the main paper. Faster vehicles increase the distance from the ego-vehicle more quickly, helping the model better perceive distant objects.
>
> | Speed  | Low  | Middle | High |
> | ------ | ---- | ------ | ---- |
> | mAP ↑  | 40.4 | 40.7   | 40.9 |
> | mATE ↓ | 64.5 | 64.2   | 63.9 |
> | mAOE ↓ | 48.3 | 48.1   | 48.0 |
> | mAVE ↓ | 27.1 | 27.1   | 27.0 |
> | NDS ↑  | 51.8 | 52.0   | 52.1 |
>
> - Asset categories: For each object category, we randomly select three assets and divide them into three groups (A/B/C ). Performance (mAP) is nearly identical across groups, indicating that visual variations have negligible effect as long as the OOD scenario is preserved.
>
> |                | car   | truck | bus   | trailer | construction vehicle | pedestrian | motorcycle | bicycle | traffic cone | barrier |
> | -------------- | ----- | ----- | ----- | ------- | -------------------- | ---------- | ---------- | ------- | ------------ | ------- |
> | Assets grout A | 0.600 | 0.354 | 0.341 | 0.106   | 0.135                | 0.468      | 0.402      | 0.411   | 0.626        | 0.565   |
> | Assets grout B | 0.594 | 0.350 | 0.343 | 0.110   | 0.135                | 0.471      | 0.405      | 0.414   | 0.625        | 0.563   |
> | Assets grout C | 0.601 | 0.352 | 0.341 | 0.108   | 0.133                | 0.470      | 0.401      | 0.411   | 0.626        | 0.565   |
>
> The speed-ablation and asset-type ablation results are added to the Appendix A.4.

---

> ### Author Response · Authors · 2025-11-27
> **Sorry to bother you**
>
> Dear reviewer dArx,
>
> Thanks for the time and effort you've put into reviewing our submission. We have provided more explanations and answers to your questions.
>
> As the discussion deadline is approaching, we kindly invite you to share your thoughts and engage in discussion. If there are any points you'd like us to clarify or discuss further, we are more than happy to assist.
>
> Thank you again for your valuable feedback and contributions.
>
> Best regards,
>
> Authors of Paper 971

---

### Author Response · Authors · 2025-11-20
**General Response 1/2**

We would like to express our sincere gratitude to all the reviewers for their thoughtful and constructive feedback on our submission. We are delighted that our paper was recognized for **“identifying a key shortcoming in most generative simulators for autonomous driving data”** (reviewers dArx, yb3h, yxwm), **“high efficiency in data augmentation”** (reviewers dArx, FTf2, yb3h), **“3D-aware guidance maps enabling realistic, geometry-consistent, multi-view scene synthesis”** (reviewer FTf2), **“DriveObj3D fills a gap in open 3D asset resources for driving research”** (reviewers dArx, FTf2, yb3h), and **“comprehensive and convincing experiments”** (reviewer yxwm).

------

**We have revised our paper (with changes highlighted in blue). The updates are summarized as follows:**

1. (For Reviewers dArx, FTf2, yb3h, yxwm). We have added a detailed description of the entire synthesis process for the 420 samples in Appendix A.2, including how the process is automated and the amount of manual effort involved.
2. (For Reviewers dArx, yxwm). We have included additional ablation studies on scene insertion, trajectory insertion, and asset-category insertion in Appendix A.4.
3. (For Reviewers yb3h, yxwm). We have added experiments on scaling the amount of OOD data in Appendix A.3.
4. (For Reviewers FTf2, yb3h, yxwm). We have added quantitative evaluations using DriveObj3D metrics in Appendix A.3.
5. (For Reviewer FTf2). We have streamlined the Introduction section, added more motivational connections in the Method section, adjusted the layout of Figure 4, and included additional relevant references.

------

**Regarding the reviewers' comments, we provide the following general responses.**

> **GR1.** Detailed synthesis procedure and time-cost analysis for generating the 420 samples.

We begin by clarifying the construction of the 420 samples, then present a detailed, step-by-step account of the time required, and conclude by describing the operations carried out at each stage.

- Construction of the 420 samples.

  We randomly select seven annotated scenes and insert ten object categories (one asset per category). Each video contains 33 frames, from which we extract six key frames for downstream training, resulting in a total of 7 × 10 × 6 = 420 samples.

- Detailed synthesis steps and time analysis.

  - |                             Step                             | Automation（s） | Manual effort（s） |
    | :----------------------------------------------------------: | --------------- | ------------------ |
    | Scene selection & initial position annotation (MLLM-assisted) | 700×10          |                    |
    |                      Scene verification                      |                 | 7×20               |
    |                 Scene orientation annotation                 |                 | 7×15               |
    |                       Asset selection                        |                 | 10×5               |
    |                    Video generation (DiT)                    | 70×10×4         |                    |
    |                            Total                             | 9800s ≈ 2.72h   | 295s               |

  - Scene selection and initial position annotation.

    An MLLM identifies scenes containing at least one collision-free straight-driving trajectory (front, rear, left, or right). For eligible scenes, it annotates the asset’s initial insertion position in ego-vehicle coordinates for the first frame (x, y, z). The full prompt is provided in Appendix A.2.

    The nuScenes training set contains 700 videos, and each MLLM query takes about 10 seconds, totaling 700 × 10 seconds.

  - Human verification of MLLM-filtered scenes.

    Only seven scenes are needed. We manually verify the MLLM-generated initial positions, requiring ~20 seconds per scene, totaling 7×20 seconds.

  - Scene orientation annotation.

    Orientation is calibrated using *pyrender* python package. For each eligible scene, we render an initial asset-insertion using nuScenes intrinsics and extrinsics to determine the correct yaw angle. The first rendering takes 6 seconds, the correction ~3 seconds, and the second rendering another 6 seconds, for a total of approximately 15 seconds per scene. Because all assets are standardized, each scene only needs to be calibrated once, totaling 7×15 seconds.

  - Asset selection.

    We require only one asset per object category, with no additional constraints. A random choice per category suffices (~5 seconds each), totaling 10×5 seconds.

  - Video generation.

    We generate 70 videos for the 420 samples. Each video uses 10 inference steps; on an H200 GPU each step takes ~4 seconds, totaling 70 × 10 × 4 seconds.

  - **Summary.**

    We adopt a partially automated pipeline. The full synthesis process for all 420 samples takes under 3 hours, with **approximately** **300 seconds of actual human labor**, which is effectively negligible.

---

> ### Author Response · Authors · 2025-11-20
> **General Response 2/2**
>
> > **GR2.** Contribution and technical novelty.
>
> - Our contribution is twofold: (1) we discover the **unfair evaluation protocol** commonly used in synthetic-data–augmented world models, and (2) we introduce a **principled augmentation strategy** that yields clear downstream gains. The novelty lies in critical insight into effective synthetic data utilization.  Notably, with **<2%** synthetic data, our method achieves substantial improvements, whereas prior work synthesizes 100% of the dataset yet shows negligible or negative impact.
>   - Prior studies [2,3] train hybrid (real + synthetic) models for twice the epochs of real-only baselines. Under a fair, epoch-matched setting, their reported gains disappear or turn negative.
>   - This exposes a key issue: exiting evaluation focuses on FID/FVD rather than on whether synthetic data improves real-world perception. High generative scores are **irrelevant** if they do not yield downstream benefits.
>   - When optimizing for downstream perception, we find that models require **distribution-shifting** samples, **not synthetic replicas of real scenes**. Simple imitation provides no benefit, whereas inserting diverse assets into real scenes creates meaningful OOD data that consistently improves performance.
>   - To our knowledge, we provide the first fair evidence that synthetic data can deliver **real advantages** over real-only training.
> - We further introduce **3D-aware guidance maps** as a new control signal for driving world models, enabling precise and flexible 3D scene editing.
>   - Existing world models [2,3,4,5] rely on 3D bounding boxes or BEV maps, but these signals often produce **position offsets, missing objects, or incorrect shapes and orientations** during object insertion [1,6,7]. Training downstream models on synthetic data with such misalignments can even harm performance.
>   - To preserve the original background while reliably inserting new assets, we propose 3D-aware guidance maps, where depth, normals, and edges maintain **background geometry**, and object patches and masks ensure accurate **foreground rendering and placement**.
>   - This representation is naturally suited for driving scene editing: 3D assets can be placed with precise coordinates and orientation, and the edited guidance maps consistently guide the DiT model to generate coherent videos.
>
> ------
>
> [1] GenMM: Geometrically and Temporally Consistent Multimodal Data Generation for Video and LiDAR. Arxiv 2024.
>
> [2] Panacea: Panoramic and Controllable Video Generation for Autonomous Driving. CVPR 2024.
>
> [3] SubjectDrive: Scaling Generative Data in Autonomous Driving via Subject Control. AAAI 2025.
>
> [4] MagicDrive-V2: High-Resolution Long Video Generation for Autonomous Driving with Adaptive Control. ICCV 2025.
>
> [5] DriveEditor: A Unified 3D Information-Guided Framework for Controllable Object Editing in Driving Scenes. AAAI 2025.
>
> [6] Realistic and Controllable 3D Gaussian-Guided Object Editing for Driving Video Generation. Arxiv 2025.
>
> [7] Vehiclesim: realistic and 3D-aware video editing with one image for autonomous driving. Multimedia Systems 2025.

---

### Author Response · Authors · 2025-11-25

Dear Reviewers,

As the discussion period is coming to a close, we want to check back to see whether you have any remaining questions. **We would be happy to clarify further, and grateful for any other feedback you may provide.**

We really appreciate your time engaged in the review and rebuttal phase.

Thank you very much and look forward to your replies!

Best regards,

Authors of Paper 971

---

### Meta-Review · Area_Chair_uyoA · 2026-01-03

**Summary:**

This paper primarily investigates the limitations of existing generation methods in downstream tasks, demonstrating that significant improvements can be achieved using only 2% of generated data. It also introduces DriveObj3D, a large-scale 3D asset dataset covering typical categories in driving scenarios and enabling diverse 3D-aware video editing.
The main concerns are lacked sufficient automation, insertion details, writting problem, missing related works, limited novelty, cost analysis. A rebuttal is provided to address most of these concerns. I am leaning to accept this paper. Author should revise the paper according to discussion.

**Reviewer Concerns:**

Concerns of all reviewers are addressed in the rebuttal.

**Reviewer Scores:**

Reviewer dArx would not change their score.
Reviewer FTf2 would not change their score.
Reviewer yb3h may change their score to 6.
Reviewer yxwm may change their score to 6.

---

### Decision · Program_Chairs · 2026-01-26

Accept (Poster)